# Frequency tunable magnetostatic wave filters with zero static power magnetic biasing circuitry

Xingyu Du [1], Mohamad Hossein Idjadi [1], Yixiao Ding[1], Tao Zhang [1], Alexander J. Geers [1], Shun Yao[1], Jun Beom Pyo[1], Firooz Aflatouni [1], Mark Allen[1] & Roy H. Olsson III [1] ✉

A single tunable filter simplifies complexity, reduces insertion loss, and minimizes size compared to frequency switchable filter banks commonly used for radio frequency (RF) band selection. Magnetostatic wave (MSW) filters stand out for their wide, continuous frequency tuning and high-quality factor. However, MSW filters employing electromagnets for tuning consume excessive power and space, unsuitable for consumer wireless applications. Here, we demonstrate miniature and high selectivity MSW tunable filters with zero static power consumption, occupying less than 2 cc. The center frequency is continuously tunable from 3.4 GHz to 11.1 GHz via current pulses of submillisecond duration applied to a small and nonvolatile magnetic bias assembly. This assembly is limited in the area over which it can achieve a large and uniform magnetic field, necessitating filters realized from small resonant cavities micromachined in thin films of Yttrium Iron Garnet. Filter insertion loss of 3.2 dB to 5.1 dB and out-of-band third order input intercept point greater than 41 dBm are achieved. The filter's broad frequency range, compact size, low insertion loss, high out-of-band linearity, and zero static power consumption are essential for protecting RF transceivers from interference, thus facilitating their use in mobile applications like IoT and 6 G networks.

The growth of multi-band and high frequency communication systems has resulted in single bandpass filter technologies being unable to satisfy the filtering requirements for all bands. This challenge arises from the congestion in the radio-frequency (RF) spectrum, encompassing the electromagnetic frequencies employed in wireless communication[1–3]. This is especially problematic as frequencies are scaled beyond the spectrum allocated for 5 G (3–6 GHz), where RF silicon-on-insulator switches exhibit unacceptably high loss when utilized in switched filter banks[4,5]. For example, the FR3 band from 7.125 to 24.25 GHz under exploration for 6 G networks will require extensive innovations in both RF switch[4,6] and acoustic filter[7–10] technologies if it is to adopt the massively parallel switched acoustic filter banks utilized in 4 G and 5 G networks. As compared to the traditional

implementation of a switched filter-bank, a single tunable filter has great potential to reduce the system cost, size, complexity, and remove entirely the additional switch paths loss[2,11–13]. Switched filter banks, commonly employed in communication systems, utilize multiple fixed filters that can be selectively activated or deactivated using RF switches to filter signals at various frequency bands. Tunable bandpass filters are needed in applications beyond 6 G wireless such as cognitive radios[14], frequency hopped receivers[15], satellite communications[16], base stations[17], and multiband radar[18].

For frequencies spanning from S band to X band (2–12 GHz), numerous tunable filters have been developed to eliminate out-of-band noise while preserving the in-band signal. However, most of these filters have a limited frequency tuning range. This is also elaborated on

[1]Department of Electrical and Systems Engineering, University of Pennsylvania, Philadelphia, PA, USA. ✉e-mail: rolsson@seas.upenn.edu

in Supplementary Note 1. Mechanically tunable filters offer high quality factor (Q) but require external circuitry and motors for tuning and have a limited center frequency range with a maximum center frequency tuning ratio of 1.2:1[19–21]. Radio Frequency Micro-Electro-Mechanical Systems (RF-MEMS) enabled tunable electromagnetic cavity filters achieve a center frequency tuning ratio up to 2:1 and high-power handling capabilities but are relatively large and sensitive to shock and vibration[22–24]. Varactor based tunable filters are small, have fast tuning speeds, and moderate center frequency tuning ratio of 2:1, but suffer from low Q values[25–28]. To the best of our knowledge, the highest reported $S_{12}$ quality factor (Q-factor) is about 46 at L band and is heavily dependent on frequency[26]. This limits the minimum filter bandwidth and the steepness of the filter skirts. In addition, while a tunable filter attenuates out-of-band interferer signals, the intermodulation distortion of the filter could permit out-of-band signals a path to mix into the filter passband which can deteriorate receiver signal-to-noise ratio. In a linear filter system, the principle of superposition dictates that the response to a combination of multiple inputs is the sum of the responses to each individual input. However, nonlinearities in filters introduce distortion arising from intermodulation between two or more inputs, giving rise to unwanted spurious signals that degrade overall signal quality. Although much higher linearity can be achieved for mechanical and RF MEMS based tunable filters, the out-of-band (OOB) $3^{rd}$ order input referred intermodulation intercept point (IIP3) of other tunable filters is usually in the range of 23.5 dBm[29] to 27 dBm[30].

Tunable filters realized using magnetostatic wave resonators (MSWR) are a promising technology to fulfill the demands of broad and continuous frequency tuning range with high quality factor >1000[31]. Magnetostatic waves (MSW), also known as dipolar spin waves, are long wavelength spin waves, where the magnetic dipolar interactions dominate both electric and exchange interactions[32]. Because the MSW group velocities are slower than that of electromagnetic waves and are variable with applied bias magnetic field, magnetic field tunable magnetostatic wave filters (MSWF) with a wide frequency tuning range are possible[32,33]. Micrometer thick, single crystal yttrium iron garnet (YIG) thin films exhibit the lowest damping for MSW and thus the smallest propagation loss as compared to other common ferromagnetic materials[34]. As a result, previous studies have demonstrated YIG based MSWR with large quality factors of 3600 at 9 GHz[35] and 5259 at 4.77 GHz[31].

Despite the small size and high Q of YIG MSWRs, YIG tunable MSWFs still suffer from large size, high power consumption, and slow tuning speed from the use of bulky and energy intensive electromagnets to supply the necessary magnetic bias field for MSWFs[36]. SI Note 1 shows a size comparison between this study and commercial YIG-based tunable filters. These filters, incorporating electromagnet drivers for YIG sphere resonator frequency tuning, result in sizes exceeding 23 cm³ and power consumption surpassing 2 W. These limitations hinder their applicability in Internet of Things (IoT) and mobile phone technologies.

Filters based on YIG sphere resonators have demonstrated low loss across a broad tuning frequency range (9:1). YIG sphere resonators, however, are too large to fit within the miniature magnetic bias circuit reported here[37,38]. Previously reported planar geometry MSWR formed through standard microfabrication processes have a form factor compatible with the reported small, tunable, magnetic bias circuits. Insertion loss is a measure of how much the filter attenuates a signal at a given frequency. Numerically, the insertion loss of a filter is the ratio of the signal level at the input of the filter to the signal level at the output of the filter. Bandstop filters have different loss tradeoffs and low loss bandstop filters have recently been reported in thin film YIG[39,40]. However, bandpass filters realized from MSWR exhibited 20 to 32 dB insertion loss when operating with a wide frequency tuning range between 2 to 12 GHz[41]. This is mainly due to the difficulty of

obtaining large coupling and a well matched impedance across a broad frequency range. Low insertion loss planar YIG tunable filters were only demonstrated over a limited frequency range: ~5.3 dB loss with a center frequency tuning ratio of 1.5:1 in X band[42] and 5.8–6.4 dB loss with a tuning ratio of 1.5:1 in X and $K_u$ bands[43]. In order to meet the requirements of insertion loss and out-of-band suppression, a planar MSWR with a large area of 4 mm × 10 mm[44] or a five layer stack MSWR with dimensions of 2 mm × 2 mm × 0.62 mm were needed[43].

In this study, we demonstrate miniature, narrowband, frequency tunable filters (3.3:1) with zero static power consumption and exceptional out-of-band linearity. Figure 1a–c depicts the device assembly and shows images of the tunable filter assembly with a total volume of only 1.68 cm³. To tune the cavity center frequency, current pulses were applied to AlNiCo pieces in the magnetic bias assembly to alter their nonvolatile magnetic remanence. Using this approach, the tunable magnetic bias circuit only consumes transient power to tune the magnetic field and filter frequency and enables the frequency tunable filter to operate without any steady state power consumption. Magnetostatic surface waves were utilized, where an in-plane magnetic bias field is established in the YIG perpendicular to the direction of magnetostatic wave propagation. The MSWF is based on cavities microfabricated in a YIG thin film with straight edge reflectors. By using planar microfabricated YIG cavities to form MSWFs, the size remains small such that the filters can fit within in the small, tunable bias assembly, which can only produce large and uniform magnetic fields over a small area. To optimize for low insertion loss, aluminum input and output transducers were placed directly on the YIG film to efficiently excite and collect the magnetostatic waves with high coupling. The geometrical parameters were also optimized based on the equivalent circuit shown in Fig. 1d to enable impedance matching to 50 Ω over the broad 3.3:1 frequency tuning range. These innovations enabled low filter insertion loss of 3.2–5.1 dB across the entire 3.4–11.1 GHz frequency tuning range with a YIG filter occupying only 200 × 70 μm² of area.

## Magnetostatic wave resonator (MSWR)

Straight edge MSWR consisting of a ferrimagnetic resonant cavity made of a 3.3 μm film of YIG grown on top of a Gadolinium Gallium Garnet (GGG) substrate were patterned into a rectangular shape by wet etching. The transducers made of 2 μm thick aluminum (Al) microstrips were fabricated on top of the YIG. The transducers width is approximately 7 μm unless stated otherwise. Figure 1c shows one typical fabricated device, where the width (W) is defined as the coupling length of the Al electromagnetic transducer on the MSW, whereas the length (L) is defined as the MSW cavity length in the direction of MSSW propagation. The fabricated MSWR was first measured under the magnetic probe station where two electromagnets were used to generate the magnetic bias field, as further illustrated in Supplementary Notes 2 and 3.

Inside the YIG cavity, the MSW is stimulated by inductive antennas, inducing oscillating magnetic fields through RF currents, as shown in Fig. 1d. The structure of the YIG cavity consists of two parallel reflecting interfaces formed by wet-etched YIG edges. As a result, spin waves entering the YIG cavity circulate coherently with minimal damping. By placing a single Al transducer on top of the YIG, the YIG cavity is configured as a MSWR. Alternatively, the MSRF employing two Al transducers produces a filter-like bandpass frequency response with high out-of-band rejection. The cross-section view depicts the unidirectional propagation of the MSW, which exclusively propagate along the surface of the YIG and are reflected onto the other surface[45,46]. As a result, the propagation path is not reciprocal between the two ports. This MSWR response can be represented by an equivalent circuit model of a parallel RLC circuit in series with an ohmic series resistance ($R_s$) and a self-inductance ($L_s$)[31]. The resonance tank within the MSWR exhibits maximum impedance at the resonance

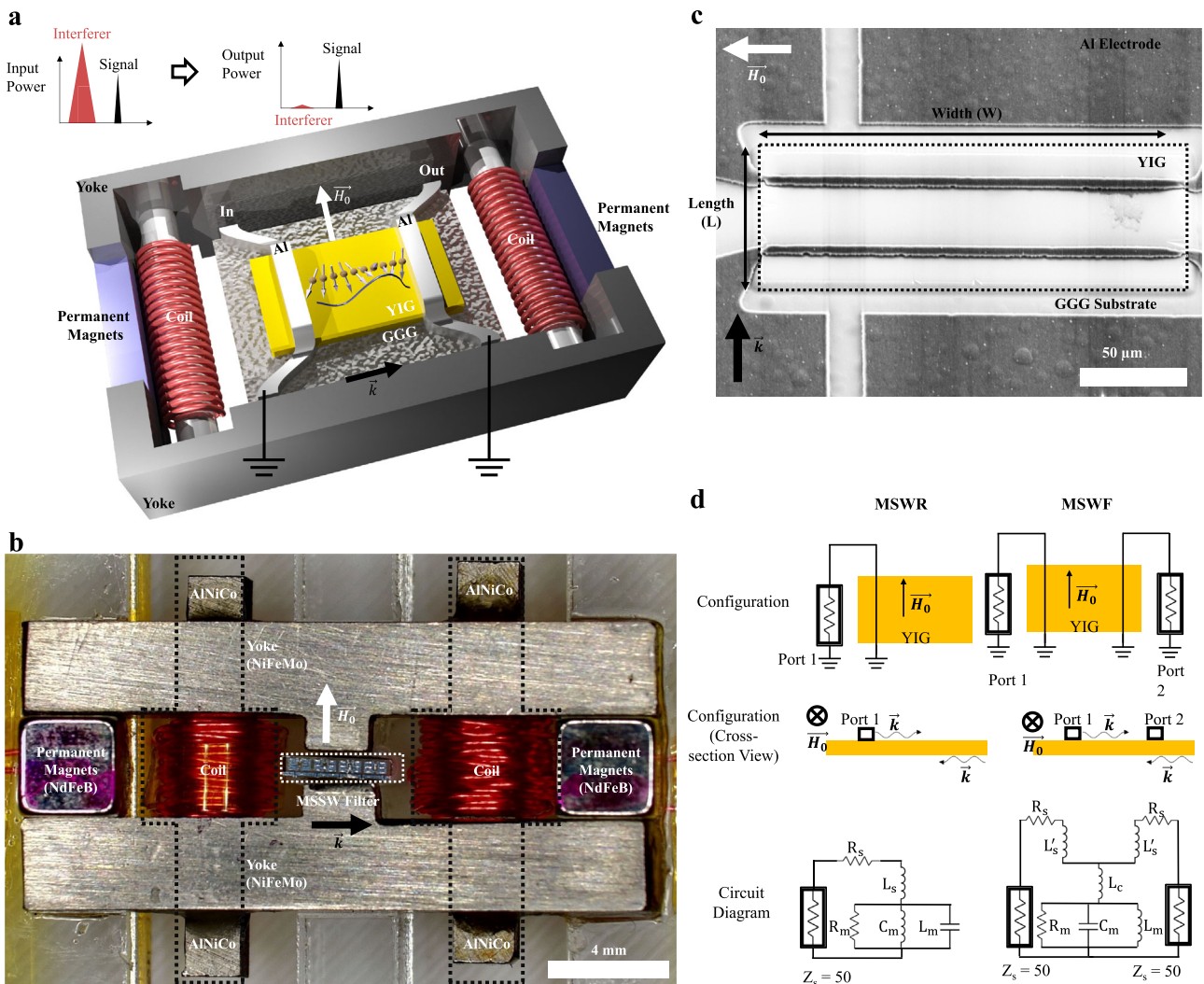

**Fig. 1 | Tunable bandpass filter with magnetic biasing circuit.** The magnetostatic wave filters (MSWF) were placed in the center of the magnetic biasing component. The aluminum transducers were placed on top of the yttrium iron garnet (YIG) cavity. The magnetic biasing component consists of two permanent magnets, two shunt magnets wrapped with coils, and two magnetically permeable yokes which concentrate the magnetic flux in the MSWF. **a** Reconfigurable MSWF concept: The primary function of a radio-frequency (RF) filter is to selectively allow certain frequencies to pass while blocking others. With the implementation of a suitable RF filter, the amplitude of the out-of-band interfering signal is significantly reduced compared to the desired signal. This feature is particularly important in scenarios where the interference is much larger than the intended signal at the receive antenna. Moreover, the high out-of-band linearity of our filter plays a vital role in ensuring that intermodulation products generated by interfering signals do not adversely impact the desired signal. **b** Optical microscope image of the fabricated device assembly. **c** Scanning Electron Microscope image showing the aluminum transducers on top of the YIG cavity. This device has a width (W) of 200 μm and length (L) of 70 μm. **d** Summary of device schematic diagrams and equivalent single-mode circuit models of MSWF and magnetostatic wave resonator (MSWR).

frequency. Consequently, the return loss displays a dip at the resonance frequency. Two-port MSWF can be modeled by connecting the resonance tank with a coupling inductor $L_c$, which considers the direct inductive coupling between the two ports where the series inductance satisfies the relationship $L_s = L_s' + L_c$.

A single RLC tank circuit with magnetostatic resistor $R_m$, magnetostatic inductor $L_m$, magnetostatic capacitor $C_m$ only captures the response at one frequency. The complete impedance response can be modeled by the multi-mode circuit model in Fig. 2a with the number of modes, p. Figure 2b compares the modeled and measured MSWR input impedance of a MSWR with W = 200 μm and L = 70 μm with an Al transducer width of 4 μm. At the MSWR's series resonance ($f_s$), also referred to as the resonance frequency, the device impedance reaches a maximum value equal to the magnetostatic resistance, $R_m$, as the parallel combination of $L_m$ and $C_m$ resonates yielding an open circuit. At $f_p$, the fundamental mode's impedance is minimal, and is often obscured by the higher impedance contribution of other modes. This anti-resonance occurs when the magnetostatic capacitor's impedance, $C_m$, is the complex conjugate of the combined impedance of the magnetostatic and series inductors, $L_m$ and $L_s$, and thus the overall impedance reaches a minimum. Moreover, the multi-mode circuit successfully predicts the impedance response across a wide frequency range. Supplementary Note 4 details the circuit modeling procedure. The circuit modeling and analysis were performed directly on the measured data without any de-embedding process.

The single-mode circuit model has proven to be effective in accurately predicting bandwidth, magnitude, and phase of the impedance around the peak frequency of the MSWR. A MSWF's performance can be optimized by increasing the coupling coefficient ($K^2$), Q-factor, and figure of merit (FoM = $K^2Q$) of the MSWR mode. The coupling coefficient, $K^2$, is the ratio of the energy stored in the magnetostatic wave, modeled by $L_m$, to the energy stored in the inductance of the transducer, modeled by $L_s$, well below the device resonance frequency, so that the energy stored in $L_m$ is not increased via resonance

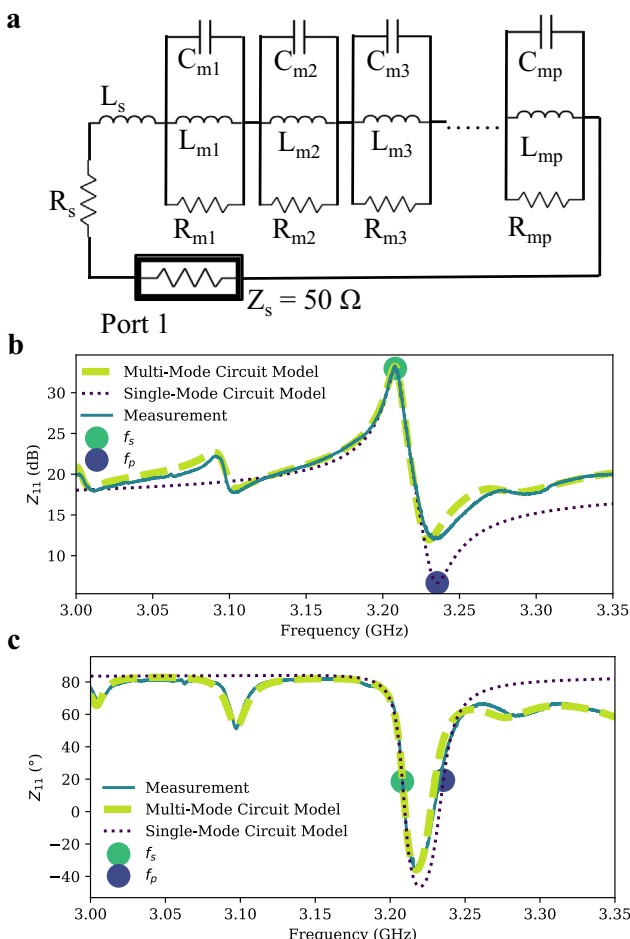

**Fig. 2 | Circuit modeling for magnetostatic wave resonator (MSWR). a** Multi-mode circuit model for MSWR. **b** Comparison of the impedance magnitude of the measured MSWR, single mode circuit model, and multi-mode circuit model. **c** Comparison of impedance phase of the measured MSWR, single mode circuit model, and multi-mode circuit model. The series resonance frequency, $f_s$ and parallel resonance frequency, $f_p$, of the single-mode circuit model have been labeled. This MSWR is with W = 200 μm and L = 70 μm with Al transducer width of 4 μm. The device is measured at an applied bias field of 500 Gauss.

with $C_m$. It can be defined as $K^2 = \frac{f_p^2 - f_s^2}{f_s^2}$ and describes how efficiently energy can be converted between electrical and magnetostatic waves by the transducers. The Q-factor is the ratio of energy stored in the MSWR, to the total energy dissipated within it, calculated as $Q = \frac{f_s}{\triangle f_s}$, where $\triangle f_s$ is the full width at half maximum (FWHM) of the impedance at the series resonance frequency. The FoM describes the contrast in the resonator impedance observed at the series resonance frequency to that observed far from resonance and is crucial for assessing the trade-off between the minimum filter insertion loss and the filter isolation when using MSWRs to form filters[47]. Fig. 3 compares the influence of the width of the MSWR on these parameters, with a fixed MSWR length of 70 μm. Increasing the width of the YIG cavity results in a higher $K^2$, FOM, and magnetostatic resistance, $R_m$, as the coupling distance increases. A maximum $K^2 = 2.4$ % is achieved at a width of 600 μm. The Q-factor does not show a significant change with the width of the YIG cavity. However, the Q-factor generally increases with frequency as previously reported[48,49]. This helps to achieve an almost constant filter bandwidth with center frequency tuning, which is one of the advantages of MSSW filters to achieve constant data rates at various frequencies. The maximum Q-factor measured is 1313, which is for

the MSWR with width of 150 μm at a frequency of 11.6 GHz. The increase in Q with frequency can be attributed to the increase in $R_m$ with frequency, while the $R_s$ remains constant. This decrease in the relative energy dissipated by the electrical transducer resistance leads to a higher Q value. Similarly, the drop in coupling with frequency is caused by the decrease in $L_m$ with frequency, while the $L_s$ remains constant. Since the $K^2$ is determined by the ratio of $L_m$ to $L_s$, a decrease in $L_m$ leads to a decrease in coupling. The high Q-factor demonstrates the excellent selectivity and efficiency of the MSWR in achieving narrow bandwidths and minimizing signal losses. The plot of magnetostatic resistance, $R_m$, vs. width shows a linearly increasing relationship. This agrees with previous theoretical calculations of the radiation resistance increase with the coupling distance[50,51]. In a simple microstrip model, the transducer self-inductance and self-resistance should be linearly proportional to its length. However, in practical scenarios, factors such as contact resistance and the resistance and inductance associated with the transducer routing introduce complexities. As a result, the total measured series resistance and inductance do not scale directly with the width of the device. For instance, the total series resistance is approximately 0.84 Ω and 2.2 Ω, and the total series inductance is 0.22 nH and 0.68 nH for W = 150 μm and 600 μm resonators respectively.

The series resistance and inductances increase by 2.6–3.1x, where a 4x scaling is expected. This suggests significant unwanted series components arise from factors such as contact resistance or other electrical routing. Future studies could explore the use of thicker aluminum transducers or new layout designs, which could increase the coupling and quality factor of the devices.

To design an MSWR with better FoM, the width effect of the Al transducers and the length effect of MSWR are also discussed in Supplementary Notes 5 and 6. Impedance matching plays a crucial role in the design of a low loss filter, as further illustrated in Supplementary Note 7. Overall, W = 150-200 μm MSWR are better matched to the 50 Ω source impedance at high frequencies whereas the W = 600 μm MSWR are matched to 50 Ω at low frequency. Although higher $K^2$ and FoM were achieved in the wide MSWR, the impedance mismatch causes the insertion loss for W = 600 μm to be higher than that of W = 100 or 200 μm when taken across the tunable frequency range.

## Magnetostatic wave filter (MSWF)

Supplementary Note 8 illustrates the tunability of the MSWF via applied magnetic field. The relationship of the main resonance frequency with respect to the applied external magnetic bias field is linear with a slope of 2.9 MHz/Gauss. Figure 4 shows the typical $S_{12}$ frequency responses of the MSWF. All these MSSW filters exhibited less than 10 dB insertion loss with greater than 20 dB out-of-band isolation.

As was discussed in the previous section, the FoM increases with increasing width, resulting in a lower insertion loss for the wide MSWF at 3.4 GHz. However, the insertion loss of W = 600 μm at 9.1 GHz is not lower than the W = 200 μm filter. This is because the main resonant tank becomes over-coupled to the source impedance of 50 Ω, leading to significant reflection loss. The length of the MSSW filter, as explained in the supplementary notes, does not have a significant effect on the FoM and $R_m$. Hence, the insertion losses of the L = 70 μm and 140 μm devices are similar.

In the filter passband, the filter response is dominated by the excitation and reception of magnetostatic waves by the input and output transducers. These transducers, which are metal lines on top of the YIG, also have a direct magnetic coupling akin to coupled inductors which dominate the response far from the filter passband and determines the filter out-of-band rejection. This direct magnetic coupling is influenced by the size of the transducers and the proximity of the input and output transducers, with longer transducers and closer proximity increasing the coupling and leading to diminished

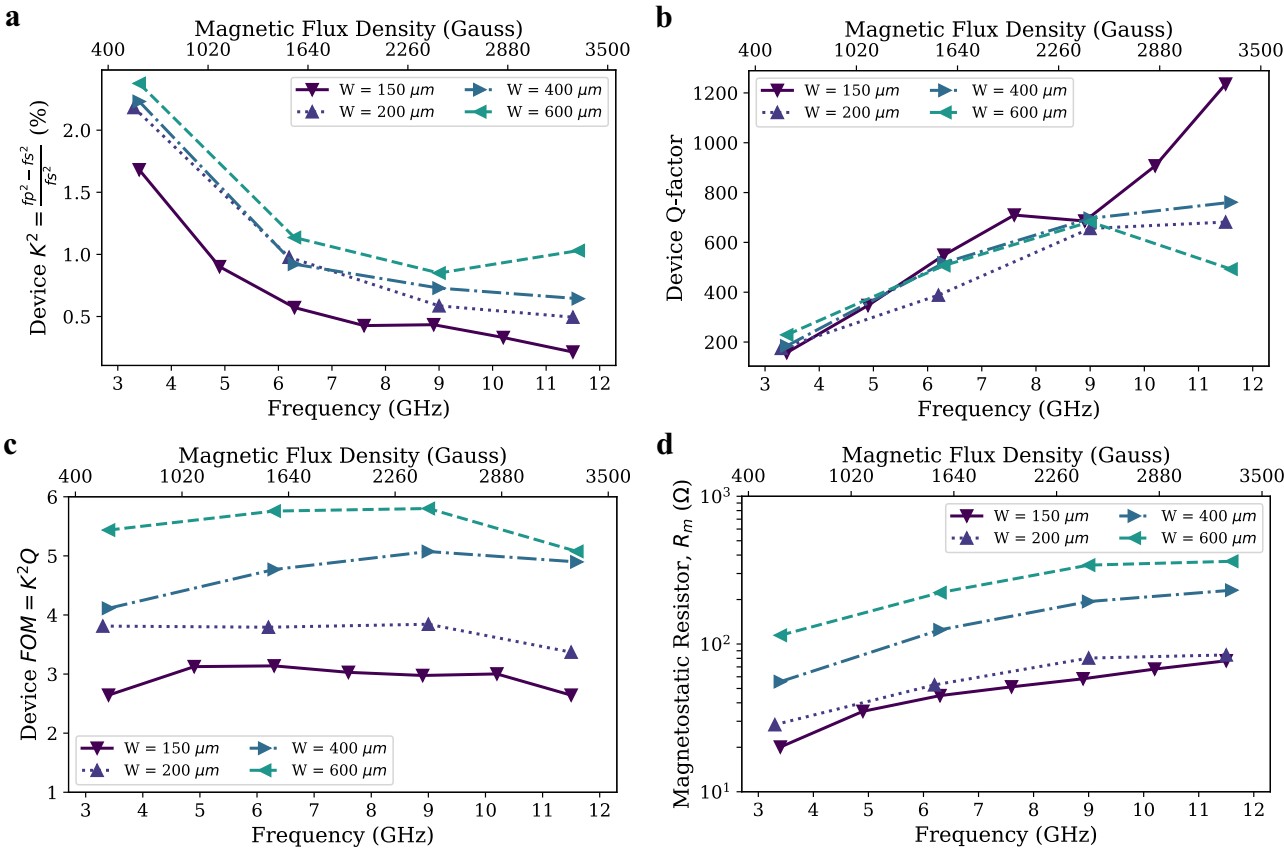

**Fig. 3 | Comparison of magnetostatic wave resonator (MSWR) for various YIG cavity widths.** The effect of yttrium iron garnet (YIG) cavity width on (**a**), the device coupling coefficient, (**b**), the device Quality factor (Q-factor), (**c**), device figure of merit (FoM), (**d**), the magnetostatic resistance, $R_m$.

out-of-band rejection in the filter. While wider widths of YIG, corresponding to longer Al transducers, result in increased coupling to the MSW, they also lead to higher direct magnetic coupling between port one and port two. It is worth noting that the direct inductive coupling can be mitigated to some extent by increasing the propagation distance of the MSW. This increased propagation distance helps to reduce the direct inductive coupling between the ports, thereby improving the out-of-band rejection performance of the MSWF.

Previous studies have shown the dispersion relations for ferrimagnetic films with finite dimensions[52]. The finite film width of ferrimagnetic films introduces a multitude of propagating modes known as width modes[52,53]. MSSW propagate exclusively along the surfaces of the YIG film and undergo reflection to the other surface at the straight edges. This reflection process gives rise to the formation of circulating wave patterns within the film. These patterns result in resonance when the round-trip phase of the MSWs inside the cavity equals 2π. It is important to note that MSWs exhibit a strongly dispersive nature, causing the resonance modes at 2π, 4π, 6π, etc. to be closely spaced in frequency. This contrasts with cavities that utilize low dispersion waves such as electromagnetic and acoustic waves. Further details on the calculation of length and width modes can be found in Supplementary Note 9. Due to the unique dispersion relationship, the W = 600 μm MSWF results in a smaller spacing between two adjacent width modes, due to an increase in the dispersion of the wider device. Additionally, an increase in width results in a frequency shift for the main resonance mode, as observed in the dispersion curve where a constant wavenumber corresponds to a higher frequency as the width increases. This frequency shift is further supported by the measurement results, which demonstrate that narrow MSWF exhibits a lower resonance frequency for the fundamental mode. Furthermore, it is evident from the calculations and measurements that MSW becomes

more dispersive at higher frequencies in wider MSWF. A longer YIG cavity showed more spurious responses as more main resonant peaks were observed in the device with L = 105 and 140 μm than the device with W = 150 μm, L = 70 μm. Supplementary Note 11 compares the differences between $S_{12}$ and $S_{21}$.

Supplementary Notes 12, 13 describe the measurement of power handling capability of the MSWF with W = 150 μm and L = 70 μm. Out-of-band signals that are outside the limits-imposed by the MSW dispersion relationship, are unable to excite MSW waves, and thus are unaffected by increases in input power. In-band $S_{12}$ decreases with the increase of the input power beyond −20 dBm at all the four measured frequencies while the out-of-band response remains unchanged. The in-band, input 1 dB compression point is −17 dBm (min) at 3.4 GHz and −14 dBm (max) at 8.9 GHz. The in-band, output 1 dB compression point exhibits a selective limiting effect associated with MSW propagation in YIG[54]. When the input power is above $P_{1dB}$ and below approximately 8 dBm, the insertion loss is increasing, and the output power is saturating with increasing input power. When the input power is above 9 dBm, the direct inductive coupling between the input and output aluminum transducers dominates the insertion loss and the output once again increases linearly with input power. Such self-limiting behavior can be useful to protect receivers from damage under large in-band interference.

Supplementary Note 13 shows the in-band and out-of-band IIP3 measurements for W = 150 μm and L = 70 μm. The in-band IIP3 does not change significantly with the resonance frequency but is a strong function of tone spacing. At a tone spacing of 1 MHz, the IIP3 displays the minimum value of −8 dBm at 7.6 GHz and −11 dBm at 10.1 GHz. The filter 3 dB bandwidth is approximately 18 to 25 MHz. The intermodulation products can be filtered at large tone spacing. Therefore, when the two-tone spacing increases to 30 MHz, the IIP3 increases to

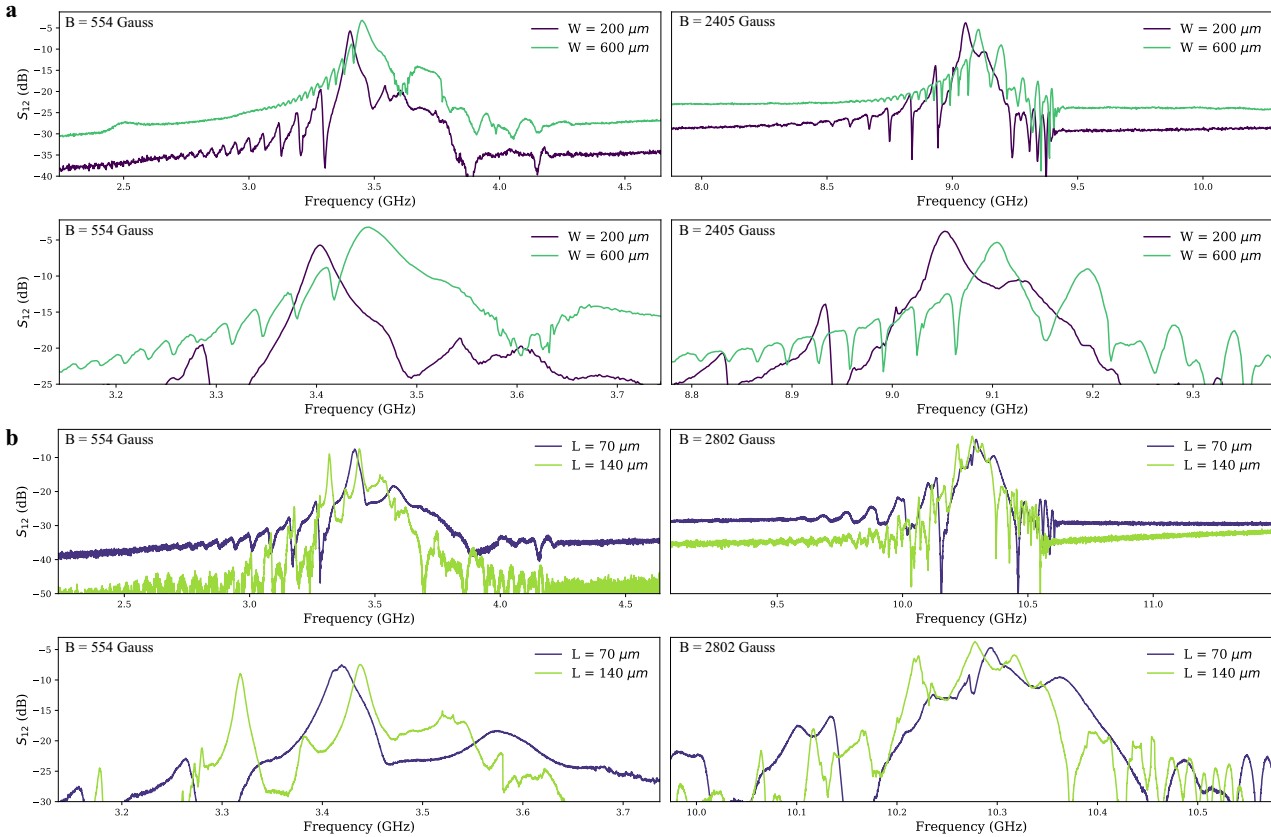

**Fig. 4 | S$_{12}$ Frequency response of the magnetostatic wave filters (MSWF) at different magnetic flux density (B). a** The impact of yttrium iron garnet (YIG) cavity width on the frequency response with a constant length of 70 μm. **b** The influence of YIG cavity length on the frequency response with a constant width of 150 μm.

8 dBm and 5 dBm at frequencies of 7.6 GHz and 10.1 GHz, respectively. Later, the MSWF was biased with a constant magnetic field of 1460 Gauss which corresponds to a resonant frequency of 6.2 GHz. The out-of-band IIP3 was measured at 5.0, 7.6, and 10.3 GHz.

Since MSW are not able to be excited outside of the allowable frequency range, the MSWF are unable to create output intermodulation products via the YIG MSW waves. The measured IIP3 is 43 dBm, 41 dBm, and 44 dBm at 5.0, 7.6, and 10.3 GHz respectively and are independent of tone spacing. This large IIP3 value highlights the resilience of this tunable filter to a strong out-of-band blocker. Out-of-band IIP was also measured with the bias magnetic field tuned to zero gauss. The IIP3 value was still in the range of 41 to 45 dBm at frequencies of 5.0, 7.6, and 10.3 GHz. This implies that the MSSW is not the primary source of out-of-band intermodulation products. Future studies with an improved IIP3 test setup or improved linearity in the aluminum transducers could achieve an increase in the out-of-band IIP3 of the tunable filters.

## Magnetic biasing circuit

As shown in Fig. 1, the magnetic bias circuit comprises two neodymium-iron-boron (NdFeB) permanent magnets, two AlNiCo magnets wrapped with copper coils and two nickel-iron-molybdenum (NiFeMo) magnetic yokes. The NiFeMo magnetic yokes provide a low reluctance path for magnetic flux due to low coercivity and high permeability. The NdFeB permanent magnets and the coil-wound AlNiCo magnets serve as a constant magnetic flux source and a tunable magnetic flux source, respectively. Compared with NdFeB material, AlNiCo material has a lower coercivity[55]. Therefore, the AlNiCo magnets can be magnetized and demagnetized by applying a pulse of current though the coils surrounding the AlNiCo material. Also, due to the high magnetic remanence of AlNiCo, the AlNiCo magnets can still

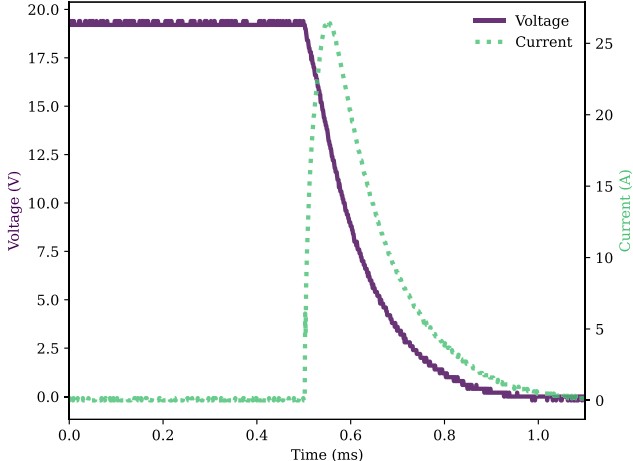

**Fig. 5 | Capacitor voltage and coil current during capacitor discharge.** During the capacitor discharge, a current pulse was applied to the coil to magnetize/demagnetize the AlNiCo magnets.

retain magnetism and provide magnetic flux for the circuit after the end of the current pulse.

To generate a pulse of current for magnetizing/demagnetizing the AlNiCo magnets, a capacitor was used. Initially, the capacitor was charged to a voltage chosen to produce the desired magnetic field. Subsequently, it was connected to the coil and discharged, which produced a current response according to a series RLC circuit. Figure 5 shows the measured voltage across the capacitor and current flowing through the coil during the capacitor discharge. The capacitor used in

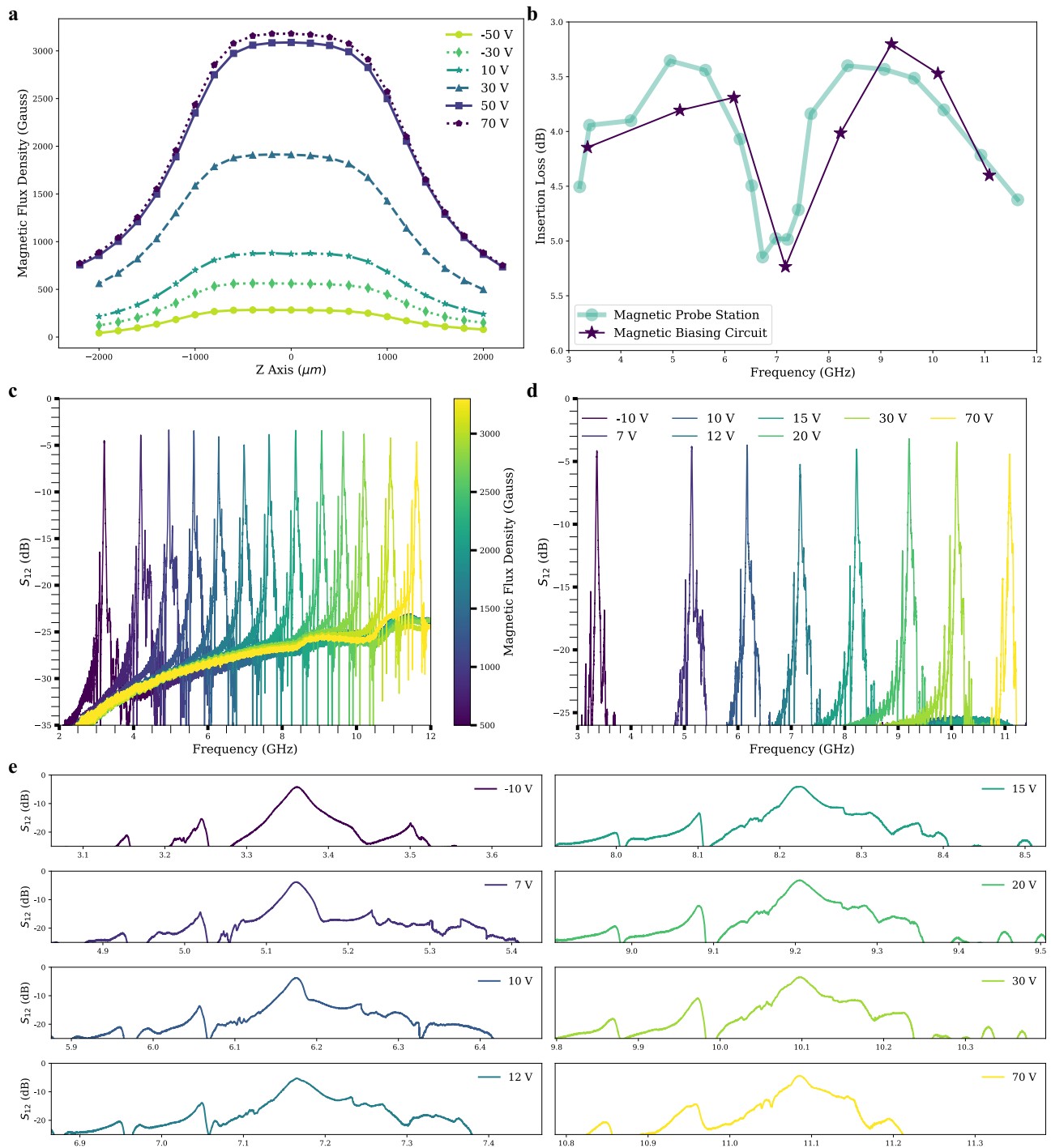

**Fig. 6 | Integrated device. a** Measured magnetic flux density under different capacitor charging voltages. **b** Insertion loss vs. frequency comparison for the magnetostatic wave filters (MSWF) measured under an external magnetic field generated by an electromagnet and the magnetic biasing circuit. **c** MSWF response measured with the magnetic field supplied by an electromagnet on a magnetic probe station. **d** MSWF response measured with the magnetic field supplied by the zero DC power tunable magnetic biasing circuit. **e** Zoomed in $S_{12}$ frequency response of MSWF measured with magnetic field supplied by the tunable magnetic biasing circuit.

the experiment had a capacitance of 270 µF and was charged to 19 V. The peak current flowing through the coil was approximately 27 A. The amplitude of the current pulse is proportional to the capacitor charging voltage. By controlling the capacitor charging voltage and the current pulse amplitude, the remanent flux of the AlNiCo magnets can be adjusted. Therefore, tuning of the magnetic field at the YIG chip is realized.

To study the tuning range of the magnetic field, the magnetic flux density in the middle of the two yokes where the YIG chip sits was measured using a Gaussmeter. By charging the capacitor to different voltages and discharging it through the coil, different current pulses were generated for magnetization. The remanent flux of the AlNiCo magnets can be adjusted by controlling the capacitor charging voltage and the resulting current pulse amplitude, allowing for precise tuning

of the magnetic field. Figure 6a shows the measured magnetic flux density along the vertical direction at different charging voltages. Supplementary Note 14 presents a comparison between the simulated and measured ranges of magnetic flux density tuning. The measured range is from 560 Gauss to 3170 Gauss, whereas the simulated range is from 450 Gauss to 3360 Gauss. The origin of the vertical position is where the magnetic flux density reaches the maximum value, which is around the middle of the yoke. Due to the 2 mm thickness of the magnetic yoke, the magnetic field does not change from −1 mm to 1 mm. This establishes a uniform magnetic bias field for the YIG filter cavity which is required for proper filter operation. Supplementary Note 15 describes a separate device in which a 0.5 mm thick yoke was utilized, resulting in a non-uniform magnetic field. This non-uniformity introduces additional losses in the integrated filter. With increasing capacitor charging voltage, the current for magnetizing the AlNiCo magnets increased, and thus the magnetic field generated by the AlNiCo magnets increased. The total amount of the magnetic flux in the middle of the two yokes was equal to the sum of the magnetic flux generated by the two NdFeB permanent magnets and the magnetic flux generated by the two AlNiCo magnets. For positive capacitor charging voltage, the field direction of the magnetized AlNiCo magnets was in the same direction as the NdFeB magnets. Due to the saturation of AlNiCo magnets, when the charging voltage was larger than 50 V, the effect of increasing the capacitor charging voltage on increasing the magnetic field became less. Finally, the tunable filter was realized by placing the MSWF chip in the center of the magnetic biasing circuit. About 0.7 J of energy and 150 μs is needed to switch from the minimum to the maximum bias field. The tunable filter assembly has a total dimension of 20 mm × 12 mm × 7 mm and occupies a volume of only 1.68 cm³.

## Integrated device

A MSWF with W = 200 μm and L = 70 μm with Al transducer width of 4 μm was laser diced and placed in the center of the gap of the magnetic biasing circuit. The filter was first measured inside a magnetic probe station with electromagnetic coils to provide the bias magnetic field. The frequency response was compared before and after integration with the tunable magnetic bias circuit. As shown in Fig. 6b–d, the $S_{12}$ frequency response and insertion loss remains unchanged between 3 GHz and 12 GHz in both the magnetic probe station and magnetic biasing circuit assemblies. The out-of-band rejection is greater than 25 dB and the insertion loss is less than 5.1 dB with an average across the tunable frequency range of 4 dB. The magnetic bias circuit measurements show a 7.7 GHz filter tuning range with a frequency tuning ratio of 3.3 achievable with an 80 V programming range. Figure 6e presents detailed $S_{12}$ frequency responses zoomed around the passband of the MSWF. The filter exhibits a clean response without significant spurious or unwanted frequency components that could interfere with the desired signal.

## Discussion

In conclusion, we have demonstrated miniature and narrowband tunable filters with zero static power consumption, exceptional out-of-band linearity, and a frequency tuning range from 3.4 to 11.1 GHz. We revealed the tradeoff in width and length of the transducers in the design of MSSW cavity resonators. Additionally, we introduced a novel bias circuit utilizing NdFeB permanent magnets, coil-wound AlNiCo magnets, and NiFeMo magnetic yokes, which allows for magnetic field tuning through applied voltage pulses. This circuit exhibits promising potential for a diverse array of applications, offering electrically controlled magnetic fields with zero static power consumption.

A filter with a YIG cavity of W = 200 μm and L = 70 μm showed the high FoM, lowest insertion loss, and largest rejection of higher order modes and out-of-band signals. This new tunable filter technology exhibits immense promise across diverse domains, notably

encompassing 5 G and 6 G cellular networks. In the realm of broadband analog-to-digital converter (ADC) technology, the tunable filter's remarkable adaptability and ability to optimize the input spectrum play a crucial role. By addressing challenges posed by wideband ADCs, such as smaller available input voltage swing and therefore reduced dynamic range, our tunable filter ensures that wideband digital receivers stay within their dynamic range and handle data efficiently, even amid changing conditions. Moreover, for broadband antennas operating at frequencies from 3 to 11 GHz, our tunable filter's compact dimensions and wide frequency tuning range represent significant breakthroughs. Integrated wideband filtering simplifies antenna designs, allowing operation across a large bandwidth while leveraging the tunable filter for selective filtering. This approach facilitates efficient coexistence of various services on the ground while reducing the need for numerous antennas and line connections[56].

The tunable filter's significance in 5 G and 6 G networks includes vital interference mitigation. For example, the sub-6 GHz 5 G spectrum can overlap with C-band VSATs for Maritime and Fixed Satellite Services. This creates unpredictable 5 G interferers, affecting users with adjacent-channel interference and Low-Noise Block (LNB) saturation[56]. Using our tunable filter in front of the LNB input can protect satellite carriers within the passband and isolate unwanted carriers, ensuring smooth operations and minimizing disruptions. The large reported out-of-band input third-order input intercept point (IIP3) of > +40 dBm allows higher levels of interfering signals before distortion. Future studies on YIG films with larger thickness can further improve the linearity of the tunable filter[54]. In addition to the band pass filter illustrated in the paper, our filter platform can readily accommodate frequency tunable notch filtering by directly connecting two ports with a transducer on top of the YIG cavity[39,40]. As illustrated in Supplementary Note 16, the MSWF will present a high impedance at resonance, effectively blocking unwanted signals at specific frequencies. This flexible and versatile design makes our tunable filter an effective device for interference management in advanced wireless networks like 5 G and 6 G.

Overall, the tunable filter's adaptability, wide frequency tuning range, low insertion loss, and zero static power consumption position it as a critical technology, effectively addressing challenges in broadband ADCs, broadband antennas, and interference mitigation in 5 G and 6 G networks. Its applications open new avenues for more efficient and dynamic RF front ends, ensuring optimal performance and seamless communication in the ever-evolving landscape of modern wireless technologies.

## Methods

### MSSW filters fabrication

The YIG was grown using liquid epitaxy on a GGG substrate with <111> orientation (prepared by MTI corporation, Ferromagnetic resonance linewidth: 0.5–2.0 Oe). A 100 nm thick $SiO_2$ was deposited as a hard mask using atomic layer deposition (Cambridge Nanotech S200) followed by 400 nm thick $SiO_2$ using plasma enhanced chemical vapor deposition (Oxford PlasmaLab 100). After annealing the sample to 600 °C in a nitrogen atmosphere, the hard mask layer was then patterned using standard photolithography and dry etching (Oxford 80 Plus RIE). The mask pattern was then transferred into the YIG using wet etching. Phosphoric acid at 140 °C was used to etch the YIG film with an etch rate of approximately 200 nm/min. The etch selectivity of YIG to $SiO_2$ using phosphoric acid was approximately 10:1. After patterning the YIG layer, the remaining $SiO_2$ layer was stripped using hydrofluoric acid. To pattern RF electrodes, 2 μm thick Al was deposited using sputtering at 1000 W with a base pressure of 1e-7 Torr (Evatec Clusterline 200 II) at 150 °C. The Al layer is patterned using standard photolithography and wet etching in Aluminum Etch Type A (Transene company, Inc) at 40 °C.

## Magnetic circuit fabrication

The three different components of the magnetic bias circuit were prepared separately and then assembled. To form the magnetic yokes, a 2 mm thick NiFeMo sheet was cut using CNC machining to the required shape as shown in Fig. 1. The main body and protrusion part of the yokes had a size of 20 mm × 3 mm and 2 mm × 1.1 mm, respectively. The NdFeB permanent magnets had a dimension of 3.175 mm × 3.175 mm × 3.175 mm and were purchased from K&J magnetics. The AlNiCo magnets had a dimension of 12 mm × 3 mm × 2 mm and were cut from a bulk AlNiCo bar using electric discharge machining (EDM). Copper wire with a diameter of 0.2 mm was wound around each AlNiCo magnet manually to achieve a total number of turns of 50. After all the magnetic parts were prepared, the two NiFeMo yokes and two NdFeB magnets were assembled and fixed on an acrylic substrate using epoxy. Then the coil-wound AlNiCo magnets were placed on the yoke and fixed using epoxy.

## Measurement setup

The YIG sample was characterized using a magnetic probe station (MicroXact's MPS-1D-5kOe). The magnetic field was generated by electromagnets inside the magnetic probe station. A Gaussmeter (Model GM2, AlphaLab Inc) was used to calibrate the magnetic probe station and the filter frequency responses were measured using a vector network analyzer (Keysight, P9374A).

## Data availability

All data supporting the findings of this study are available within the article and its supplementary files. Any additional requests for information can be directed to, and will be fulfilled by, the corresponding author.

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

## Acknowledgements

We would like to thank Dr. Timothy Hancock and Dr. David Abe of the Defense Advanced Research Projects Agency (DARPA) and Dr. Michael Page of the Air Force Research Laboratory for their guidance and support of this work under the DARPA Wideband Adaptive RF Protection (WARP) program, contract FA8650-21-1-7010. The fabrication of devices was performed at the Singh Center for Nanotechnology, supported by the NSF National Nanotechnology Coordinated Infrastructure Program (No. NNCI-1542153).

## Author contributions

X.D., M.I., F.A., M.A., and R.O. came up with the device concepts and experimental implementations. X.D., M.I., Y.D., T.Z., A.G., S.Y., J.P., F.A., M.A, and R.O. designed the devices and fabrication process flow. X.D. and M.I. fabricated the magnetostatic filters. Y.D., T.Z., and J.P. fabricated magnetic bias circuit. X.D., M.I., and S.Y. performed the measurements. X.D. and R.O. analyzed all data and wrote the manuscript. All authors have given approval to the final version of the manuscript.

## Competing interests

Provisional patent application filed.
