## [Peer Review File · Nature Communications]

Reviewers' Comments:

Reviewer #1:

Remarks to the Author:

This reviewer enjoyed reading the manuscript. The filter results are exceptional and the authors have done an excellent job comparing them to state-of-art.

While a couple of groups have recently demonstrated micro machined YIG resonators, what really stands out to this reviewer is the Magnetic Biasing Circuit and its operation. In the opinion of this reviewer SI #13 contains valuable information that should really be included in the main body of the paper since the entire design, and its application is indeed quite novel and critical to the operation of the system as zero power tunable filter.

One question that the authors may answer is what happens to the biasing circuit/programmable magnet in the presence of an EMP? or is that not relevant?

Other than this suggestion this reviewer has only two minor requests:

1. on page 7 lines 131-132, the authors state that the Q-factor generally increases with frequency. Can the authors back this up with a reference, provide their own insight or opinion about this fact instead of using the words "generally increases". Maybe they have a collection of data from multiple resonators across multiple fab runs.

the authors may also consider referencing several notch filter papers in literature on page 18 line 292.

<https://pubs.aip.org/aip/apl/article-abstract/98/21/212502/340245/Planar-millimeter-wave-band-stop-filters-based-on?redirectedFrom=fulltext>

<https://ieeexplore.ieee.org/document/10111074>

Reviewer #2:

Remarks to the Author:

The authors presented an extensive investigation of a microwave resonator/filter using the magnetostatic waves of a 3.3 μm YIG film. They obtain high tunability, low insertion loss, and large out-of-band IIP3, which might be very relevant for 5G and 6G wireless networks. Moreover, the authors assembled an integrated device that is able to miniaturize the RF filter down to a size smaller than 2 cc, a few times smaller than the current commercialized RF-MEMS.

Although the very clever miniaturization and extensive documented data, the manuscript at the current stage, has issues regarding the visual presentation, spin-wave physics concepts, and a few experimental details. Overall, all the Figures in the main text, as well as in the SI, are not vector figures and have low resolution. I understand that a higher resolution figure will be presented in the final version, but at the current stage, I was not able to read some y-axis labels. The manuscript also will benefit from breaking down a few concepts and details on the RF system in order to make the reading easier for a broad audience. After a revised manuscript correctly addresses the comments below, I can make a favourable recommendation for further consideration.

1 - Line 36: It might be useful, especially for a broad audience to specify what RF-MEMS stands for.

2 - Line 65: The authors mentioned that previous reports using MSWR exhibited 20 to 32 dB insertion loss. However, recent results also demonstrated insertion loss of 2 to 3 dB using planar geometry, Micromachined Tunable Magnetostatic Forward Volume Wave Bandstop Filter, IEEE MICROWAVE AND WIRELESS TECHNOLOGY LETTERS, VOL. 33, NO. 6, JUNE 2023. The authors are discussing a different kind of MSWR? If not, it might be important to update that information, maybe a specific literature revision is needed.

3 - Line 104: I think that should be MSRF, not MSWR.

4 - I got slightly confused with the definition of f_s , resonance frequency, and f_p , anti-resonance frequency for the resonator case, MSWR, Figure 2 b. First, was applied any magnetic field for the presented measurement? It is not indicated in the text or in the caption of the figure if there is a field applied between the poles, or if the measurement was obtained at zero field.

- If it was obtained at zero field, what does that mean a resonance at ~ 3 GHz? It is the resonance of the circuit, independent of the magnetic field, or the resonance of the YIG? If so, why there is a resonance around 3 GHz even in the absence of an external magnetic field?
- If indeed a magnetic field was applied, please indicate the value. Breaking down those details will make the reading easier to follow and digest.

5 - Figure 3: Figure 3 has a very low resolution. I understand that a higher-quality figure will be presented in a final version but at the current stage it is impossible to read the description of the y-axis in Figure 3 a). Especially because is the only place in the manuscript that defines $K^2 = (f_p^2 - f_s^2) / (f_s^2)$, I only could read the definition on Figure 10 at the supplementary material. Moreover, frequently the Q factor is calculated as $Q = \Delta f / (f_r)$, where f_r is the resonance frequency and Δf is the full width half maximum of the spectrum. The authors should indicate how they calculate the Q-factor, and also describe the difference between both. And why $FoM = K^2 Q$ is a good figure of merit.

6 - Line 170: The direct electromagnetic wave coupling, means the microwave transmission through air? The increase of the direct electromagnetic wave is a consequence of wider aluminium transducers? It was not clear in supplementary note 2 if the width of the transducers is larger for the wider YIG filters. The authors should indicate those information.

7 - Line 175: The authors mentioned the propagation modes in YIG as "width modes". I believe that the more general formalism relies on the orientations between the direction of the spin wave, or wave propagation, and the magnetization of the YIG. For the case of magnetization in the plane of the film, there are two modes, magnetostatic surface-wave modes (MSSW), or Damon-Eshbach modes, when k is perpendicular to M , and backward volume spin wave modes (BVSW), when k is parallel to M , J. Appl. Phys. 128, 161101 (2020). I imagine that the authors have used the name "width modes" in order to refer to the modes that propagate along the width of the filter. However, indicating this information will facilitate the understanding of a broad audience.

8 - Figure 4: The authors mentioned the chirality of propagation of the spin waves (k). Where "positive" k propagates from the top of the surface from port 1 to port 2, while "negative" k propagates at the bottom surface from port 2 to port 1, as indicated in Figure 1 d. However, in all figures of MSWF is plotted S_{12} , which stands for measure microwave that reaches port 1 excited at port 2. That means the authors are measuring the spin wave propagating in the bottom surface of the YIG film? Moreover, there are significant intensity differences between S_{12} and S_{21} ? It might be elusive if the authors could add in the supplementary material a comparison of S_{12} and S_{21} .

9 - Minor comment: In order to tune the MSWF using the integrated device, it is needed to charge the capacitor to a specific voltage and discharge across the coils of the AlNiCo magnets. The authors can estimate the time needed to switch between two magnetic field bias? It might be interesting to know how fast this integrated system can select each frequency.

Reviewer #3:

Remarks to the Author:

The following manuscript demonstrates a series of novel magnetostatic wave tunable filters. This work is noteworthy from other magnetostatic device work since it logically demonstrates major progress in integration, magnetic tunability and filter performance. The magnetic biasing circuit shows a novel concept for providing tunable bias and allows for concentrated high fields in a small area.

To strengthen the work, the following minor revisions should be completed:

In figure 1a, the jammer/signal notching inset should be removed or further explained in the

caption or text.

In figure 4, the S_{12} of the filter out of band response is not smooth (containing ripples) which may be a result of not uniform magnetic field or volume waves (as described very briefly in the text). It is recommended that this is further addressed in the text by showing the field uniformity over the filter. It is also recommended to show S_{21} of the filter response.

In figure 5a, does the measured magnetic flux density match that of simulation? It is recommended to show a simulated bias over the specified z-axis, perhaps even include a magnetic flux density heat map overlaid on the MSSW filter image. Additionally, it is recommended that the dissimilarity of the magnetic probe station and magnetic biasing circuit be further explained and to take higher resolution data of the magnetic biasing circuit.

In figure 5d, the y-axis of s_{12} extends above 0dB.

In figure 5e, it is recommended that the y-axis should contain finer resolution to clearly show the insertion loss of the filter.

It would strengthen the manuscript if the linewidth of the YIG film used be stated in the work.

In the acknowledgement section please confirm the spelling of the names of the individuals.

REVIEWER COMMENTS

Reviewer #1 (Remarks to the Author):

This reviewer enjoyed reading the manuscript. The filter results are exceptional and the authors have done an excellent job comparing them to state-of-art. While a couple of groups have recently demonstrated micro machined YIG resonators, what really stands out to this reviewer is the Magnetic Biasing Circuit and its operation.

Answer to the reviewer:

We appreciate the recognition from the reviewer of our experimental accomplishments.

In the opinion of this reviewer SI #13 contains valuable information that should really be included in the main body of the paper since the entire design, and its application is indeed quite novel and critical to the operation of the system as zero power tunable filter.

Answer to the reviewer:

We agree with the reviewer that SI #13 which discusses the changes of Capacitor Voltage and Coil Current During Capacitor Discharging are important and vital to the reader. SI #13 has been moved to the main body of the paper.

Magnetic Biasing Circuit Section:

To generate a pulse of current for magnetizing/demagnetizing the AlNiCo magnets, a capacitor was used. Initially, the capacitor was charged to a voltage chosen to produce the desired magnetic field. Subsequently, it was connected to the coil and discharged, which produced a current response according to a series RLC circuit. Figure 5 shows the measured voltage across the capacitor and current flowing through the coil during the capacitor discharge. The capacitor used in the experiment had a capacitance of 270 μF and was charged to 19 V. The peak current flowing through the coil was approximately 27 A. The amplitude of the current pulse is proportional to the capacitor charging voltage. By controlling the capacitor charging voltage and the current pulse amplitude, the remanent flux of the AlNiCo magnets can be adjusted. Therefore, tuning of the magnetic field at the YIG chip is realized.

Fig. 5 in the main text: Capacitor voltage and coil current during capacitor discharge.

One question that the authors may answer is what happens to the biasing circuit/programmable magnet in the presence of an EMP? or is that not relevant?

Answer to the reviewer:

During the measurements of the devices, the authors did not observe any sudden changes of the magnetic field or demagnetization of the magnetic biasing circuit. We believe the discussion of performance in the presence of an electromagnetic pulse (EMP) is beyond the scope of this paper.

Other than this suggestion this reviewer has only two minor requests: 1. on page 7 lines 131-132, the authors state that the Q-factor generally increases with frequency. Can the authors back this up with a reference, provide their own insight or opinion about this fact instead of using the words "generally increases". Maybe they have a collection of data from multiple resonators across multiple fab runs.

Answer to the reviewer:

In response to the reviewer's suggestion, we appreciate the opportunity to enhance the clarity of our statement regarding the Q-factor and its relationship with frequency (page 7, lines 131-132).

The observed increase in Q-factor with frequency is a well-documented phenomenon in the context of Magnetostatic wave filters. This characteristic is not merely attributed to fabrication variances, as supported by findings in the literature.

In a recent paper that reported a low loss bandpass filter using YIG/GGG film structures, Shanshan et al. reported the external quality factor increase from 110 at 5 GHz to 260 at 19 GHz.¹ The reported trend aligns with our own observations. They also did not find significant changes in bandwidth with respect to resonance frequency.

In another paper that demonstrates a bandpass filter using YIG films². Their measurements and theoretical calculation agree with each other, and they demonstrated an unloaded Q-factor increase with frequency. They also observed a similar bandwidth of the tunable filter of 16~23 MHz across the resonance frequency from 0.5 ~4 GHz.

Additionally, Sen et al. explored Magnetostatic Wave YIG Resonators, discovering a peak in the Q-factor at 5 GHz, followed by a decrease at higher frequencies. Their investigation attributed this decline to unexpected high resistivity in the electrodes and contact resistance, indicating that beyond a certain bias (and resonant frequency), electrical resistance becomes a limiting factor for Q.³

Magnetostatic Wave Resonator (MSWR) Section:

However, the Q-factor generally increases with frequency as previously reported.^{1,2}

the authors may also consider referencing several notch filter papers in literature on page 18 line 292

<https://pubs.aip.org/aip/apl/article-abstract/98/21/212502/340245/Planar-millimeter-wave-band-stop-filters-based-on?redirectedFrom=fulltext>
<https://ieeexplore.ieee.org/document/10111074>

Answer to the reviewer:

We appreciate your thoughtful suggestions and contributions to improving the quality of our manuscript. These two references have been added.

Reviewer #2 (Remarks to the Author):

The authors presented an extensive investigation of a microwave resonator/filter using the magnetostatic waves of a 3.3 μm YIG film. They obtain high tunability, low insertion loss, and large out-of-band IIP3, which might be very relevant for 5G and 6G wireless networks. Moreover, the authors assembled an integrated device that is able to miniaturize the RF filter down to a size smaller than 2 cc, a few times smaller than the current commercialized RF-MEMS.

Answer to the reviewer:

We sincerely thank the reviewer for acknowledging and recognizing our efforts in presenting an extensive investigation of a microwave resonator/filter utilizing magnetostatic waves in a 3.3 μm YIG film.

Although the very clever miniaturization and extensive documented data, the manuscript at the current stage, has issues regarding the visual presentation, spin-wave physics concepts, and a few experimental details. Overall, all the Figures in the main text, as well as in the SI, are not vector figures and have low resolution. I understand that a higher resolution figure will be presented in the final version, but at the current stage, I was not able to read some y-axis labels.

Answer to the reviewer:

We appreciate the reviewer's feedback and have addressed the concerns raised regarding the visual presentation. The figures in both the main text and the Supplementary Information have been updated to Vector Figures in SVG format, ensuring higher resolution and improved clarity. We understand the importance of clear and readable figures, and these updates aim to enhance the overall quality of the manuscript. We believe that the revised figures will now allow for a more comprehensive and accessible understanding of the presented data. Your attention to detail is valuable, and we thank you for helping us improve the visual aspects of our manuscript.

The manuscript also will benefit from breaking down a few concepts and details on the RF system in order to make the reading easier for a broad audience.

Answer to the reviewer:

We appreciate the reviewer's suggestion to break down concepts and provide additional details to improve the accessibility of the manuscript for a broad audience. Following your feedback, we have incorporated explanations for the specified concepts, and we are open to further clarification if needed. The added explanations are as follows:

1. Radio-frequency (RF) spectrum: The radio-frequency (RF) spectrum refers to the range of electromagnetic frequencies used for wireless communication.

Introduction Section:

This challenge arises from the congestion in the radio-frequency (RF) spectrum, encompassing the electromagnetic frequencies employed in wireless communication.

2. Switched filter banks: Switched filter banks are structures used in communication systems to selectively filter signals at different frequency bands.

Introduction Section:

Switched filter banks, commonly employed in communication systems, utilize multiple fixed filters that can be selectively activated or deactivated using RF switches to filter signals at various frequency bands.

3. Insertion loss: Insertion loss is a measure of how much the filter attenuates a signal at a given frequency. Numerically, the insertion loss of a filter is the ratio of the signal level at the input of the filter to the signal level at the output of the filter.

Introduction Section:

Insertion loss is a measure of how much the filter attenuates a signal at a given frequency. Numerically, the insertion loss of a filter is the ratio of the signal level at the input of the filter to the signal level at the output of the filter.

4. Linear filter: A linear filter is a filter whose output is a linear combination of its input values. In other words, the response of a linear filter to a weighted sum of input values is equal to the weighted sum of the responses to each individual input value.

Introduction Section:

In a linear filter system, the principle of superposition dictates that the response to a combination of multiple inputs is the sum of the responses to each individual input. However, non-linearities in filters introduce distortion arising from intermodulation between two or more inputs, giving rise to unwanted spurious signals that degrade overall signal quality.

We believe that these additions enhance the clarity of the manuscript, providing readers with a better understanding of key concepts. If there are any additional concepts that require further explanation, please let us know, and we will address them accordingly. We appreciate your valuable feedback in refining our manuscript.

After a revised manuscript correctly addresses the comments below, I can make a favourable recommendation for further consideration

1 - Line 36: It might be useful, especially for a broad audience to specify what RF-MEMS stands for.

Answer to the reviewer:

We appreciate the constructive feedback provided by the reviewer. In response to the suggestion to specify the acronym RF-MEMS for the benefit of a broad audience, we have made the necessary revision. The term RF-MEMS stands for Radio Frequency Micro-Electro-Mechanical Systems, and this clarification has been added to the manuscript.

Introduction Section:

Radio Frequency Micro-Electro-Mechanical Systems (RF-MEMS) enabled tunable electromagnetic cavity filters achieve a center frequency tuning ratio up to 2:1 and high-power handling capabilities but are relatively large and sensitive to shock and vibration.^{4, 5, 6}

2 - Line 65: The authors mentioned that previous reports using MSWR exhibited 20 to 32 dB insertion loss. However, recent results also demonstrated insertion loss of 2 to 3 dB using planar

geometry, Micromachined Tunable Magnetostatic Forward Volume Wave Bandstop Filter, IEEE MICROWAVE AND WIRELESS TECHNOLOGY LETTERS, VOL. 33, NO. 6, JUNE 2023. The authors are discussing a different kind of MSWR? If not, it might be important to update that information, maybe a specific literature revision is needed.

Answer to the reviewer:

We appreciate your suggestion and the reference to the recent paper describing Micromachined Tunable Magnetostatic Forward Volume Wave Bandstop Filter. In Line 65, we were referring to bandpass filters. However, the suggested paper is a bandstop filter. Bandpass and bandstop filters serve different purposes in the RF front end.

In the paragraph mentioned (Line 65), we are specifically addressing the limitations of bandpass filters using MSWR. The insertion loss discussed pertains to the challenges faced in achieving low insertion loss for bandpass filters due to factors such as magnetostatic wave propagation loss, electrical loss, and coupling loss. The difficulty lies in simultaneously achieving low propagation loss, high coupling coefficient, and high quality factor.

The referenced paper describes a bandstop filter, which operates differently from a bandpass filter. In a bandstop filter, the goal is to block the signal at the resonance frequency. In such filters, the insertion loss is primarily limited by the electrical loss of the transducers since there is no magnetostatic wave in the passband.

These distinctions make achieving low insertion loss in a bandstop filter considerably easier compared to a bandpass filter. To ensure clarity on this point, we have incorporated the following sentence in the main text of our manuscript:

Introduction Section:

Bandstop filters have different loss tradeoffs and low loss bandstop filters have recently been reported in thin film YIG.^{7,8} However, bandpass filters realized from MSWR exhibited 20 to 32 dB insertion loss when operating with a wide frequency tuning range between 2 to 12 GHz.⁹

3 - Line 104: I think that should be MSRF, not MSWR.

Answer to the reviewer:

We appreciate your keen observation. We acknowledge the mistake in using MSWR instead of MSRF. We thank you for bringing this to our attention. The manuscript has been revised accordingly.

Introduction Section:

Alternatively, the **MSRF** employing two AI transducers produces a filter-like bandpass frequency response with high out-of-band rejection.

4 - I got slightly confused with the definition of f_s , resonance frequency, and f_p , anti-resonance frequency for the resonator case, MSWR, Figure 2 b. First, was applied any magnetic field for the presented measurement? It is not indicated in the text or in the caption of the figure if there is a field applied between the poles, or if the measurement was obtained at zero field.

- If it was obtained at zero field, what does that mean a resonance at ~3 GHz? It is the resonance of the circuit, independent of the magnetic field, or the resonance of the YIG? If so, why there is a resonance around 3 GHz even in the absence of an external magnetic field?
- If indeed a magnetic field was applied, please indicate the value. Breaking down those details will make the reading easier to follow and digest.

Answer to the reviewer:

We appreciate this finding. There is indeed a magnetic field applied in Figure 2b. Otherwise the MSWR won't exist. We've updated the Figure 2 caption to indicate that the device was measured with an applied magnetic field of 500 Gauss. This addition should clarify the observed resonance at approximately 3 GHz.

Fig. 2:

The device is measured at an applied bias field of 500 Gauss.

The resonance frequency f_s and anti-resonance frequency f_p are foundational in the Modified Butterworth Van-Dyke (MBVD) model for microresonators, as discussed in previous literature^{10, 11}. These concepts are now being utilized to evaluate the performance of the MSWR. At these two frequencies, the phase angle is zero and the resonator impedance is purely real. However, the primary distinction between the MSWR and the MBVD model lies in the configuration of the resonance components. While the MSWR uses a parallel arrangement of $R_m L_m C_m$, the MBVD model connects these components in series.

In the MSWR, the series resonance f_s , also known as resonance frequency, occurs when the magnetostatic inductor's impedance is the complex conjugate of the magnetostatic capacitor (i.e. equal in absolute value and 180 degrees opposite in phase), leading to the equation:

$$f_s = \frac{1}{2\pi\sqrt{L_m C_m}}$$

At resonance, the parallel LC tank circuit has the impedance of an open circuit, with the circuit impedance mainly determined by the magnetostatic resistance, R_m . As a result, the total impedance at resonance in a MSWR is essentially the magnetostatic resistance, R_m resulting in high resistance and low current at f_s . The mathematical expression for f_s is the same as in the MBVD model. However, in the MBVD model, the series resonance f_s defines the frequency which exhibits minimum impedance.

The parallel resonance, f_p , also known as the anti-resonance frequency, occurs when the magnetostatic capacitor matches that of the combined magnetostatic inductor, L_m , and series inductor, L_s , leading to

$$f_p = \frac{1}{2\pi} \sqrt{\frac{L_m + L_s}{C_m L_m L_s}} = f_s \sqrt{\frac{L_m + L_s}{L_s}}$$

At this point, the overall impedance reaches the minimum equal to R_s . This can also be expressed as

$$f_p = f_s \sqrt{1 + K_{eff}^2}$$

Where K_{eff}^2 is defined as

$$K_{eff}^2 = \frac{f_p^2 - f_s^2}{f_p^2} = \frac{L_m}{L_s} = \frac{K^2}{1 + K^2}$$

The definition of effective coupling coefficient K_{eff}^2 and the coupling coefficient K^2 , are the same as those in acoustic resonators.¹²

Given the above explanation, the following sentences have been added to the manuscript.

Magnetostatic Wave Resonator (MSWR) Section:

At the MSWR's series resonance (f_s), also referred to as the resonance frequency, the device impedance reaches a maximum value equal to the magnetostatic resistance, R_m , as the parallel combination of L_m and C_m resonates yielding an open circuit.

At f_p , the fundamental mode's impedance is minimal, and is often obscured by the higher impedance contributions of other modes. This anti-resonance occurs when the magnetostatic capacitor's impedance, C_m , is the complex conjugate of the combined impedance of the magnetostatic and series inductors, L_m and L_s , and thus the overall impedance reaches a minimum.

5 - Figure 3: Figure 3 has a very low resolution. I understand that a higher-quality figure will be presented in a final version but at the current stage it is impossible to read the description of the y-axis in Figure 3 a). Especially because is the only place in the manuscript that defines $K^2 = (f_p^2 - f_s^2) / (f_s^2)$, I only could read the definition on Figure 10 at the supplementary material. Moreover, frequently the Q factor is calculated as $Q = \Delta f / (f_r)$, where f_r is the resonance frequency and Δf is the full width half maximum of the spectrum. The authors should indicate how they calculate the Q-factor, and also describe the difference between both. And why $FoM = K^2 Q$ is a good figure of merit.

Answer to the reviewer:

We appreciate the reviewer's suggestion. We have taken steps to improve the quality of Figure 3 to ensure better readability.

Moreover, the detailed explanation of the K^2 and Q factor have been added to the manuscript. The coupling coefficient, K^2 , is the ratio of the energy stored in the magnetostatic wave, modeled by L_m , to the energy stored in the inductance of the transducer, modeled by L_s , well below the device resonance frequency, so that the energy stored in L_m is not increased via resonance with C_m . It can be defined as $K^2 = \frac{f_p^2 - f_s^2}{f_s^2}$ and describes how efficiently energy can be converted between electrical and magnetostatic wave energy by the transducers. The Q-factor, in contrast, is the ratio of energy stored in the MSWR to the total energy dissipated within it, calculated as $Q = \frac{f_s}{\Delta f_s}$, where Δf_s is the full width at half maximum (FWHM) of the impedance at the series resonance frequency.

Lastly, the figure of merit, $FoM = K^2 Q$, has been widely used in acoustic resonator studies. It describes the contrast in the resonator impedance observed at the resonance frequency to that observed far from resonance, which serves as an important parameter for assessing the tradeoff between minimum filter insertion loss and filter isolation.¹² The following explanation has been added to the manuscript:

Magnetostatic Wave Resonator (MSWR) Section:

The coupling coefficient, K^2 , is the ratio of the energy stored in the magnetostatic wave, modeled by L_m , to the energy stored in the inductance of the transducer, modeled by L_s , well below the device resonance frequency, so that the energy stored in L_m is not increased via resonance with C_m . It can be defined as $K^2 = \frac{f_p^2 - f_s^2}{f_s^2}$ and describes how efficiently energy can be converted between electrical and magnetostatic waves by the transducers. The Q-factor is the ratio of energy stored in the MSWR, to the total energy dissipated within it, calculated as $Q = \frac{f_s}{\Delta f_s}$ where Δf_s is the full width at half maximum (FWHM) of the impedance at the series resonance frequency. The FoM describes the contrast in the resonator impedance observed at the series resonance frequency to that observed far from resonance and is crucial for assessing the trade-off between the minimum filter insertion loss and the filter isolation when using MSWRs to form filters.¹²

6 - Line 170: The direct electromagnetic wave coupling, means the microwave transmission through air? The increase of the direct electromagnetic wave is a consequence of wider aluminum transducers? It was not clear in supplementary note 2 if the width of the transducers is larger for the wider YIG filters. The authors should indicate those information.

Answer to the reviewer:

We appreciate the reviewer's questions and clarifications regarding the direct electromagnetic wave coupling and the width of the YIG filters.

The magnetostatic transducers' desired function is to excite magnetostatic waves by creating a magnetic field in the YIG by passing an RF current through the input transducer or to receive magnetostatic waves, which will create an RF current in the output transducer in response to a magnetostatic wave. However, when this electric current passes through the input transducer conductor, it also creates a magnetic field outside the YIG directly at the output transducer. This time varying magnetic field induces a current in the output transducer conductor, a principle akin to transformers. This induction results in an undesired direct coupled current in the output transducer (i.e. a current at the output that is not from the magnetostatic wave), thereby diminishing the filter's out-of-band rejection.

The strength of this inductive coupling is significantly affected by the size of the transducer and the proximity of the input and output transducers. Specifically, longer transducers produce stronger magnetic fields, which in turn strengthens the direct coupling effect. Consequently, it was inferred in the manuscript that a wider YIG, which has longer transducers, typically leads to reduced out-of-band rejection. However, we apologize for any confusion: the term "wider" was intended to describe the width of the YIG cavity, not the aluminum transducer. In Fig. 1c, the length of the aluminum transducer is the same as the width of the YIG cavity.

We have incorporated these clarifications into the manuscript to improve understanding. Thank you for pointing out these areas that needed further explanation.

Magnetostatic Wave Filter (MSWF) Section:

In the filter passband, the filter response is dominated by the excitation and reception of magnetostatic waves by the input and output transducers. These transducers, which are metal lines on top of the YIG, also have a direct magnetic coupling akin to coupled inductors which dominates the response far from the filter passband and determines the filter out-of-band rejection. This direct magnetic coupling is influenced by the size of the transducers and the proximity of the input and output transducers, with longer transducers and closer proximity increasing the coupling leading to diminished out-of-band rejection in the filter. While wider widths of YIG, corresponding to longer Al transducers, result in increased coupling to the MSW, they also lead to higher direct magnetic coupling between port one and port two.

7 - Line 175: The authors mentioned the propagation modes in YIG as “width modes”. I believe that the more general formalism relies on the orientations between the direction of the spin wave, or wave propagation, and the magnetization of the YIG. For the case of magnetization in the plane of the film, there are two modes, magnetostatic surface-wave modes (MSSW), or Damon-Eshbach modes, when k is perpendicular to M , and backward volume spin wave modes (BVSW), when k is parallel to M , J. Appl. Phys. 128, 161101 (2020). I imagine that the authors have used the name “width modes” in order to refer to the modes that propagate along the width of the filter. However, indicating this information will facilitate the understanding of a broad audience.

Answer to the reviewer:

We greatly appreciate your insightful observation regarding the various categories of dipolar spin waves, including magnetostatic surface waves, forward volume waves, and backward volume waves. In our manuscript, the alignment of the magnetic field with the propagation direction corresponds specifically to magnetostatic surface waves (MSSW). Consequently, it was these MSSW that were predominantly observed in our experiments.

Your comment also highlights an important parallel between light propagation in optical waveguides and magnetostatic wave propagation in magnetostatic devices. In optical waveguides, such as fibers, light behavior is governed by the waveguide's geometry and material properties, leading to the formation of various modes. These modes include Transverse Electric (TE), Transverse Magnetic (TM), as well as Transverse Electric and Magnetic (TEM) modes, and their higher order variants, each characterized by distinct electromagnetic field distribution patterns.

Similarly, in the context of magnetostatic surface wave resonators, various modes can emerge even within the same MSSW category. These modes are influenced by the finite dimensions of the sample, particularly its width. Modes resulting from the sample's width are termed 'width modes' in prior literature on YIG resonators¹³, so we have also used that terminology. If the Yttrium Iron Garnet (YIG) sample is infinitely wide ($w \rightarrow \infty$), the magnetostatic waves would resemble plane waves, rendering these width modes indistinguishable.

To provide a comprehensive understanding of this phenomenon, Supplementary Note 9 in our manuscript offers an extensive analysis of the modes in Magnetostatic Wave Resonators. This analysis elucidates how the finite dimensions of the resonators give rise to distinct mode patterns, akin to the behavior observed in optical waveguides.

We hope this clarification addresses your query and adds depth to our study's findings regarding spin wave behavior in magnetostatic surface wave resonators. The following sentence has been added to Supplementary Note 9.

Supplementary Note 9: Mode analysis of Magnetostatic Wave Resonators (MSWR)

Although a wider MSWR can provide higher FoM and lower insertion loss, it also contains additional spurious responses akin to the response of a multimode waveguide associated with the width of the YIG structure.

8 - Figure 4: The authors mentioned the chirality of propagation of the spin waves (k). Where “positive” k propagates from the top of the surface from port 1 to port 2, while “negative” k propagates at the bottom surface from port 2 to port 1, as indicated in Figure 1 d. However, in all figures of MSWF is plotted S_{12} , which stands for measure microwave that reaches port 1 excited at port 2. That means the authors are measuring the spin wave propagating in the bottom surface of the YIG film? Moreover, there are significant intensity differences between S_{12} and S_{21} ? It might be elusive if the authors could add in the supplementary material a comparison of S_{12} and S_{21} .

Answer to the reviewer:

Thank you for your question regarding the propagation of magnetostatic waves and the nonreciprocity of the filter in our study. As depicted in Fig. 1d, the MSWs within the Yttrium Iron Garnet (YIG) cavity are excited by input inductive antenna transducers. The YIG cavity's structure plays a crucial role in this process. It consists of two parallel reflecting interfaces, created by the wet-etching of YIG edges. This configuration allows spin waves to enter and circulate within the YIG cavity coherently, with minimal energy loss due to low damping. Essentially, the magnetostatic waves oscillate multiple times inside the YIG cavity before being collected by the output transducer. Thus, standing waves corresponding to both S_{12} and S_{21} circulate around the YIG cavity hundreds to thousands of times, depending on Q , propagating on both the bottom and top surfaces as they pass between the ports.

The primary difference between S_{12} and S_{21} lies in the initial point of interaction within the oscillating loop. For S_{21} , representing the signal journey from port 1 to port 2, the signal initially propagates away from port 1 along the top surface, next it interacts with the port 2 aluminum transducer, then reflects to the bottom YIG surface at the reflecting YIG edge, then propagates along the bottom YIG surface, reflects back to the top YIG surface at the other reflecting YIG edge, and arrives back at port 1 before repeating this cycle. Conversely, S_{12} , signifying the signal path from port 2 to port 1, first encounters the two YIG edges before interacting with the port 1 aluminum transducer. Its path is as follows: the signal initially propagates away from port 2 along the top surface, then reflects to the bottom YIG surface at the reflecting YIG edge, then propagates along the bottom YIG surface, then reflects back to the top YIG surface at the other reflecting YIG edge before arriving at port 1. From port 1 it propagates along the top surface to port 2 before repeating this cycle.

This variance in the signal path leads to a slight nonreciprocal behavior in the filter's performance. As a result, S_{12} and S_{21} exhibit minor differences. To illustrate this, we have included an example

in Supplementary Note 11. This note demonstrates that while S_{12} and S_{21} have similar insertion losses, S_{12} shows marginally better performance in suppressing spurious modes.

It's important to note that in some applications, such as in the RF front-end following the receiving antenna, only one-way signal propagation is typically required. The underlying causes of these slight dissimilarities between S_{12} and S_{21} are the subject of our ongoing research and are considered beyond the scope of this current manuscript.

Supplementary Note 11: Comparison of S_{12} and S_{21}

Supplementary Figure 19. Examples of comparison of the S_{12} and S_{21} . The MSWF with $W = 200 \mu\text{m}$ and $L = 70 \mu\text{m}$ with Al transducer width of $4 \mu\text{m}$. The MSWF is bias with magnetic flux density of 790 Gauss.

9 - Minor comment: In order to tune the MSWF using the integrated device, it is needed to charge the capacitor to a specific voltage and discharge across the coils of the AlNiCo magnets. The authors can estimate the time needed to switch between two magnetic field bias? It might be interesting to know how fast this integrated system can select each frequency.

Answer to the reviewer:

We appreciate the reviewer's insightful question regarding the time required to switch between two magnetic field bias settings in our integrated system.

In response to your question, we have added a paragraph to the main text addressing the charging process of the magnetic biasing circuit. Figure 5 now displays the measured voltage across the capacitor and the current flowing through the coil during the capacitor discharging process.

Based on our measurements, the magnetic bias circuit starts to charge at 0.5 ms and completes the charging process at 0.65 ms. Overall, it takes approximately 150 μs to switch between different magnetic field settings.

Magnetic Biasing Circuit Section:

To generate a pulse of current for magnetizing/demagnetizing the AlNiCo magnets, a capacitor was used. Initially, the capacitor was charged to a voltage chosen to produce the desired magnetic field. Subsequently, it was connected to the coil and discharged, which produced a current response according to a series RLC circuit. Fig. 5 shows the measured voltage across the capacitor and current flowing through the coil during the capacitor discharge. The capacitor used in the experiment had a capacitance of 270 μF and was charged to 19 V. The peak current flowing through the coil was approximately 27 A. The amplitude of the current pulse is proportional to the capacitor charging voltage. By controlling the capacitor charging voltage and the current pulse amplitude, the remanent flux of the AlNiCo magnets can be adjusted. Therefore, tuning of the magnetic field at the YIG chip is realized. Approximately 0.7 J and 150 μs is needed to switch from the minimum to the maximum bias field.

Fig. 5 in the main text: Capacitor voltage and coil current during capacitor discharge.

Reviewer #3 (Remarks to the Author):

The following manuscript demonstrates a series of novel magnetostatic wave tunable filters. This work is noteworthy from other magnetostatic device work since it logically demonstrates major progress in integration, magnetic tunability and filter performance. The magnetic biasing circuit shows a novel concept for providing tunable bias and allows for concentrated high fields in a small area.

Answer to the reviewer:

We sincerely appreciate the positive feedback and recognition from the reviewer regarding our work on novel magnetostatic wave tunable filters. Your feedback is valuable and motivating as we continue to explore new frontiers in this field.

To strengthen the work, the following minor revisions should be completed: In figure 1a, the jammer/signal notching inset should be removed or further explained in the caption or text.

Answer to the reviewer:

Thank you for your question regarding the figure inset of the RF (Radio Frequency) jammer.

An RF interference can obstruct the reception of radio signals by emitting radio frequencies close to that used by the targeted communication devices. This emission creates noise or signal interference, disrupting the normal operation of these devices. In today's densely populated RF environment, interferers may emit signals more powerful than the desired communication signals, or they may be more powerful when received at the communications device due to the proximity to the desired and interfering transmitters.

The primary function of an RF filter is to selectively allow certain frequencies to pass while blocking others. With the implementation of a suitable RF filter, the amplitude of the interfering signal is significantly reduced compared to the desired signal. This feature is particularly important in scenarios where the interference is much larger than the intended signal. Moreover, the high out-of-band linearity of our proposed filter plays a vital role in ensuring that intermodulation products generated by interfering signals do not adversely impact the wanted signal.

The following sentences have been added.

Fig. 1 Tunable Bandpass Filter with Magnetic Biasing Circuit:

(a) Reconfigurable MSWF concept: The primary function of an RF filter is to selectively allow certain frequencies to pass while blocking others. With the implementation of a suitable RF filter, the amplitude of the out-of-band interfering signal is significantly reduced compared to the desired signal. This feature is particularly important in scenarios where the interference is much larger than the intended signal at the receive antenna. Moreover, the high out-of-band linearity of our filter plays a vital role in ensuring that intermodulation products generated by interfering signals do not adversely impact the desired signal.

In figure 4, the S12 of the filter out of band response is not smooth (containing ripples) which may

be a result of not uniform magnetic field or volume waves (as described very briefly in the text). It is recommended that this is further addressed in the text by showing the field uniformity over the filter. It is also recommended to show S21 of the filter response.

Answer to the reviewer:

We appreciate the reviewers' question regarding the out-of-band response's ripples. These ripples do not result from a non-uniform magnetic field. Supplementary Note 15 discusses the scenarios where the magnetic field is not uniform. The ripples occur because various modes coexist in the YIG cavity.

Various modes can emerge even within the magnetostatic surface wave with finite dimensions. These different modes create ripples in the out-of-band frequency response, although the magnetic field is uniform. These modes are influenced by the finite dimensions of the sample. Modes resulting from the sample's width are termed 'width modes' in prior literature.^{13, 14} If the Yttrium Iron Garnet (YIG) sample is infinitely wide ($w \rightarrow \infty$), the magnetostatic waves would resemble plane waves, rendering these width modes indistinguishable. To provide a comprehensive understanding of this phenomenon, Supplementary Note 9 in our manuscript offers an extensive analysis of the modes in Magnetostatic Wave Resonators.

These ripples are inherent to magnetostatic wave resonators and were due to the magnetostatic wave's unique dispersion relationship. Kok Wai Chang and Waguih Ishak have previously identified the resonance modes in magnetostatic surface wave resonators by numbering them according to their width and length modes.¹⁴ In a magnetostatic surface wave straight edge resonator similar to, but much larger than, that presented in the manuscript, the frequency response of this rectangular straight-edge resonator with a given dimension is a family of resonant modes for a fixed external magnetic field. For each principal resonance in this magnetostatic surface wave straight edge resonator, different odd or even width mode resonances are excited depending on the current distribution on the microstrip transducers. It is important to note that all these modes are magnetostatic surface waves, which are not related to the backward volume spin wave or forward volume spin wave.

Figure 3. Typical S_{21} frequency response of a MSSW-SER

Figure 1. Copied from figure 3 in reference paper¹⁴

S_{12} and S_{21} exhibit minor differences. To illustrate this, we have included an example in Supplementary Note 11. This note demonstrates that while S_{12} and S_{21} have similar insertion losses, S_{12} shows marginally better performance in suppressing spurious modes.

As depicted in Fig. 1d, the MSWs within the Yttrium Iron Garnet (YIG) cavity are excited by input inductive antenna transducers. The YIG cavity's structure plays a crucial role in this process. It consists of two parallel reflecting interfaces, created by the wet etching of YIG edges. This configuration allows spin waves to enter and circulate within the YIG cavity coherently, with minimal energy loss due to low damping. Essentially, the magnetostatic waves oscillate multiple times inside the YIG cavity before being collected by the output transducer. Thus, standing waves corresponding to both S_{12} and S_{21} circulate around the YIG cavity hundreds to thousands of times, depending on Q, propagating on both the bottom and top surfaces as they pass between the ports.

The primary difference between S_{12} and S_{21} lies in the initial point of interaction within the oscillating loop. For S_{21} , representing the signal journey from port 1 to port 2, the signal initially propagates away from port 1 along the top surface, next it interacts with the port 2 aluminum transducer, then reflects to the bottom YIG surface at the reflecting YIG edge, then propagates along the bottom YIG surface, reflects back to the top YIG surface at the other reflecting YIG edge, and arrives back at port 1 before repeating this cycle. Conversely, S_{12} , signifying the signal path from port 2 to port 1, first encounters the two YIG edges before interacting with the port 1 aluminum transducer. Its path is as follows: the signal initially propagates away from port 2 along the top surface, then reflects to the bottom YIG surface at the reflecting YIG edge, then propagates along the bottom YIG surface, then reflects back to the top YIG surface at the other reflecting YIG edge before arriving at port 1. From port 1 it propagates along the top surface to port 2 before repeating this cycle. This variance in the signal path leads to a slight nonreciprocal behavior in the filter's performance.

It's important to note that in some applications, such as in the RF front-end following the receiving antenna, only one-way signal propagation is typically required. The underlying causes of these slight dissimilarities between S_{12} and S_{21} are the subject of our ongoing research and are considered beyond the scope of this current manuscript.

Supplementary Note 11: Comparison of S_{12} and S_{21}

Supplementary Figure 19. Examples of comparison of the S_{12} and S_{21} . The MSWF with $W = 200 \mu\text{m}$ and $L = 70 \mu\text{m}$ with Al transducer width of $4 \mu\text{m}$. The MSWF is bias with magnetic flux density of 790 Gauss.

In figure 5a, does the measured magnetic flux density match that of simulation? It is recommended to show a simulated bias over the specified z-axis, perhaps even include a magnetic flux density heat map overlaid on the MSSW filter image.

Answer to the reviewer:

We appreciate the reviewer's question on the simulation for magnetic flux density. An Ansys simulation has been used to calculate the magnetic flux density inside the air gap between the yokes and is shown in Supplementary Note 14 below.

Magnetic Biasing Circuit Section:

Supplementary Note 14 presents a comparison between the simulated and measured ranges of magnetic flux density tuning. The measured range is from 560 Gauss to 3170 Gauss, whereas the simulated range is from 450 Gauss to 3360 Gauss.

Supplementary Note 14: Simulation for magnetic flux density

Supplementary Figure 28. Setup of the ANSYS simulation of the magnetic flux density. The two olive boxes are used to represent the NdFeB permanent magnets. The yoke (NiFeMo) is purple in color and has the same shape as the actual device. The two white colored boxes under the yoke are the two shunt AlNiCo magnets.

The coils were excluded from the simulation. This simulation encompassed two scenarios: first, the pair of shunt AlNiCo magnets with full magnetization, and second, the same magnets with no magnetization. These scenarios mimic the magnetic bias circuit at its maximum and minimum voltage applications, respectively, and serve as benchmarks for estimating the lowest and highest achievable magnetic fields.

Figure 3 shows the magnetic flux density in the YIG chip area when there is a complete magnetization of the shunt magnets. The maximum magnetic flux density is 336 mT or 3360 Gauss.

Supplementary Figure 29. (a) Depiction of the simulated magnetic flux density distribution in the central gap region. Two grey boxes, positioned at the top and bottom, symbolize the yokes. This simulation was conducted under conditions of full magnetization of the shunt magnets. (b): Illustration of the simulated magnetic flux density along the center line of the central gap area. Two boxes at the top and bottom represent the yokes. This simulation was performed assuming full magnetization of the shunt magnets. 1 mTesla is equal to 10 Gauss.

Figure 4 shows the magnetic flux density in the YIG chip area when there is a zero magnetization of the shunt magnets. The maximum magnetic flux density is 45 mT or 450 Gauss.

Supplementary Figure 30. (a) Depiction of the simulated magnetic flux density distribution in the central gap region. Two grey boxes, positioned at the top and bottom, symbolize the yokes. This simulation was conducted under conditions of zero magnetization of the shunt magnets. (b) Illustration of the simulated magnetic flux density along the center line of the central gap area. Two boxes at the top and bottom represent the yokes. This simulation was performed assuming zero magnetization of the shunt magnets. 1 mTesla is equal to 10 Gauss.

As shown in the main text Figure 6(a), The measured achieved magnetic flux density ranges from 56 mT to 317 mT or 560 to 3170 Gauss. The simulated B- magnetic flux density agrees well with the measured result. The small difference may be due to the air gaps between the shunt magnets and yokes and the air gaps between the permanent magnets and yokes, which narrows the achievable magnetic flux density range. In the simulation, all the contacts were assumed to be perfect. All the magnets and yokes were contacted intimately with no air gaps.

Additionally, it is recommended that the dissimilarity of the magnetic probe station and magnetic biasing circuit be further explained and to take higher resolution data of the magnetic biasing circuit.

Answer to the reviewer:

We appreciate the reviewer's question on the dissimilarity of the frequency response when measured with magnetic probe station versus magnetic biasing circuit. The following sentences have been added to supplementary note 14.

Supplementary Note 15:

Supplementary Figure 33 shows a scenario where the magnetic bias field is uniform, and a similar low insertion loss has been achieved in both magnetic biasing circuit and on the magnetic probe station. The magnetic biasing circuit is biased at a slight lower magnetic field, which results in the resonance frequency of 9.028 GHz being slightly lower than the resonance frequency inside the magnetic probe station. Overall, the frequency response is similar.

Supplementary Figure 33. Comparison of the frequency response of the MSWF inside the magnetic bias circuit and on the magnetic probe station where the applied magnetic field of the magnetic bias circuit is uniform. S₁₂ frequency response of the width = 150 μm, length = 70 μm MSWF. The thick yoke (2 mm thick) is used here for the magnetic biasing circuit to achieve better uniformity.

In figure 5d, the y-axis of s12 extends above 0dB.

Answer to the reviewer:

We sincerely appreciate the reviewer's keen observation regarding Figure 5d and the extension of the y-axis of s12 above 0 dB. In response to your feedback, we have made the necessary adjustment to Figure 5d. The maximum of the y-axis has been revised to 0 dB.

In figure 5e, it is recommended that the y-axis should contain finer resolution to clearly show the insertion loss of the filter.

Answer to the reviewer:

We sincerely appreciate the reviewer's feedback and recommendation to enhance the resolution of the y-axis in Figure 5e to better display the insertion loss of the filter.

In response to your suggestion, we have updated Figure 6e to provide a finer resolution on the y-axis. This adjustment ensures that the insertion loss of the filter is presented with greater clarity and accuracy.

It would strengthen the manuscript if the linewidth of the YIG film used be stated in the work.

Answer to the reviewer:

We appreciate the reviewer's suggestion to include the linewidth of the YIG film used in our work. In response to your suggestion, we have obtained the linewidth value from the manufacturer and added the following information to the manuscript:

MSSW filters fabrication section in Methods:

The YIG was grown using liquid epitaxy on a GGG substrate with <111> orientation (prepared by MTI corporation **Ferromagnetic resonance linewidth: 0.5-2.0 Oe**).

In the acknowledgement section please confirm the spelling of the names of the individuals.

Answer to the reviewer:

We appreciate your attention to detail and your concern regarding the spelling of names in the acknowledgment section. We can confirm that the spelling of the names of the individuals mentioned in the acknowledgment section now is correct. We have ensured the accuracy of their names in our manuscript.

Acknowledgement:

The authors would like to thank Dr. **Timothy Hancock** and Dr. **David Abe** of the Defense Advanced Research Projects Agency (DARPA) and Dr. Michael Page of the Air Force Research Laboratory for their guidance and support of this work under the DARPA Wideband Adaptive RF Protection (WARP) program, contract FA8650-21-1-7010. The fabrication of devices was

performed at the Singh Center for Nanotechnology, supported by the NSF National Nanotechnology Coordinated Infrastructure Program (No. NNCI-1542153).

Reference:

1. Du S, Yang Q, Fan X, Wang M, Zhang H. A Compact and Low-Loss Tunable Bandpass Filter Using YIG/GGG Film Structures. *IEEE Microwave and Wireless Technology Letters* **33**, 259-262 (2022).
2. Murakami Y, Ohgihara T, Okamoto T. A 0.5-4.0-GHz Tunable Bandpass Filter Using YIG Film Grown by LPE. *IEEE Transactions on Microwave Theory and Techniques* **35**, 1192-1198 (1987).
3. Dai S, Bhave SA, Wang R. Octave-Tunable Magnetostatic Wave YIG Resonators on a Chip. *IEEE Transactions on Ultrasonics, Ferroelectrics, and Frequency Control* **67**, 2454-2460 (2020).
4. Sinanis MD, Adhikari P, Jones TR, Abdelfattah M, Peroulis D. High-Q High Power Tunable Filters Manufactured With Injection Molding Technology. *IEEE Access* **10**, 19643-19653 (2022).
5. Liu X, Katehi LPB, Chappell WJ, Peroulis D. High-Q Tunable Microwave Cavity Resonators and Filters Using SOI-Based RF MEMS Tuners. *Journal of Microelectromechanical Systems* **19**, 774-784 (2010).
6. Yang Z, Zhang R, Peroulis D. Design and Optimization of Bidirectional Tunable MEMS All-Silicon Evanescent-Mode Cavity Filter. *IEEE Transactions on Microwave Theory and Techniques* **68**, 2398-2408 (2020).
7. Feng Y, Tiwari S, Bhave SA, Wang R. Micromachined Tunable Magnetostatic Forward Volume Wave Bandstop Filter. *IEEE Microwave and Wireless Technology Letters*, (2023).
8. Lu L, Song Y-Y, Bevivino J, Wu M. Planar millimeter wave band-stop filters based on the excitation of confined magnetostatic waves in barium hexagonal ferrite thin film strips. *Applied Physics Letters* **98**, (2011).
9. Castéra JP, Hartemann P. Magnetostatic wave resonators and oscillators. *Circuits, Systems and Signal Processing* **4**, 181-200 (1985).
10. Buttry DA, Ward MD. Measurement of interfacial processes at electrode surfaces with the electrochemical quartz crystal microbalance. *Chemical Reviews* **92**, 1355-1379 (1992).

11. Chen L, Wang Y. Dependence of modified butterworth van-dyke model parameters and magnetoimpedance on dc magnetic field for magnetoelectric composites. *Materials* **14**, 4730 (2021).
12. Nordquist CD, Olsson III RH. Radio Frequency Microelectromechanical Systems (RF MEMS). In: *Wiley Encyclopedia of Electrical and Electronics Engineering*.
13. O’Keeffe TW, Patterson RW. Magnetostatic surface-wave propagation in finite samples. *Journal of Applied Physics* **49**, 4886-4895 (1978).
14. Chang KW, Ishak W. Magnetostatic surface wave straight-edge resonators. *Circuits, systems and signal processing* **4**, 201-209 (1985).

Reviewers' Comments:

Reviewer #2:

Remarks to the Author:

The authors have correctly addressed my remarks as well as the other referees. The current version of the manuscript is well-structured and clearer to a wider audience. I commend the authors for their efforts in including additional information, such as capacitor voltage and coil current during capacitor discharge, in the supplemental material. Overall, the authors have performed admirably, and I strongly recommend the manuscript for publication in Nature Communications.

Reviewer #3:

Remarks to the Author:

The revised manuscript has been improved from the earlier submitted manuscript. The added clarifications in the paper have improved the quality of the manuscript allowing to make it much easier for the broader audience, especially the improved section on coupling and figure of merits. Furthermore, the improved graphics show the data much more clearly.

REVIEWERS' COMMENTS

Reviewer #2 (Remarks to the Author):

The authors have correctly addressed my remarks as well as the other referees. The current version of the manuscript is well-structured and clearer to a wider audience. I commend the authors for their efforts in including additional information, such as capacitor voltage and coil current during capacitor discharge, in the supplemental material. Overall, the authors have performed admirably, and I strongly recommend the manuscript for publication in Nature Communications.

Answer to the reviewer:

We appreciate the recognition from the reviewer of our accomplishments.

Reviewer #3 (Remarks to the Author):

The revised manuscript has been improved from the earlier submitted manuscript. The added clarifications in the paper have improved the quality of the manuscript allowing to make it much easier for the broader audience, especially the improved section on coupling and figure of merits. Furthermore, the improved graphics show the data much more clearly.

Answer to the reviewer:

We sincerely thank the reviewer for acknowledging and recognizing our efforts